# ENSLOSS: Stochastic Calibrated Loss Ensembles
# for Preventing Overfitting in Classification

**Ben Dai** [1]

## Abstract

Empirical risk minimization (ERM) with a computationally feasible surrogate loss is a widely accepted approach for classification. Notably, the convexity and calibration (CC) properties of a loss function ensure consistency of ERM in maximizing accuracy, thereby offering a wide range of options for surrogate losses. In this article, we propose a novel ensemble method, namely ENSLOSS, which extends the ensemble learning concept to combine loss functions within the ERM framework. A key feature of our method is the consideration on preserving the "legitimacy" of the combined losses, i.e., ensuring the CC properties. Specifically, we first transform the CC conditions of losses into loss-derivatives, thereby bypassing the need for explicit loss functions and directly generating calibrated loss-derivatives. Therefore, inspired by Dropout, ENSLOSS enables loss ensembles through one training process with doubly stochastic gradient descent (i.e., random batch samples and random calibrated loss-derivatives). We theoretically establish the statistical consistency of our approach and provide insights into its benefits. The numerical effectiveness of ENSLOSS compared to fixed loss methods is demonstrated through experiments on a broad range of 45 pairs of CIFAR10 datasets, the PCam image dataset, and 14 OpenML tabular datasets and with various deep learning architectures. Python repository and source code are available on GITHUB.

## 1. Introduction

The objective of binary classification is to categorize each instance into one of two classes. Given a feature vector $\mathbf{X} \in \mathcal{X} \subset \mathbb{R}^d$, a classification function $f : \mathbb{R}^d \to \mathbb{R}$

produces a predicted class $\text{sign}(f(\mathbf{x}))$ to predict the true class $Y \in \{-1, +1\}$. The performance of the classification function $f$ is typically evaluated using the risk function based on the zero-one loss:

$$R(f) = \mathbb{E}\big(\mathbf{1}(Y f(\mathbf{X}) \leq 0)\big), \tag{1}$$

where $\mathbf{1}(\cdot)$ is an indicator function, and the classification accuracy is defined as $\text{Acc}(f) = 1 - R(f)$. Clearly, the Bayes decision rule $f^*(\mathbf{x}) = \text{sgn}(\mathbb{P}(Y = 1 | \mathbf{X} = \mathbf{x}) - 1/2)$ is a minimizer of the risk function $R(f)$. Due to the discontinuity of the indicator function, the zero-one loss is usually replaced by a *convex* and *classification-calibrated* loss $\phi$ to facilitate the empirical computation (Lin, 2004; Zhang, 2004b; Bartlett et al., 2006). For example, typical losses including the hinge loss $\phi(z) = (1 - z)_+$ for SVMs (Cortes & Vapnik, 1995), the exponential loss $\phi(z) = \exp(-z)$ for AdaBoost (Freund & Schapire, 1995; Hastie et al., 2009), and the logistic loss $\phi(z) = \log(1 + \exp(-z))$ for logistic regression (Cox, 1958), and others (Lin et al., 2017; Leng et al., 2022). Then, the risk based on $\phi(\cdot)$ is defined as:

$$R_\phi(f) = \mathbb{E}\big(\phi(Y f(\mathbf{X}))\big).$$

Note that convexity and classification-calibration (hereafter referred to as calibration for simplicity) are widely accepted requirements for a loss function $\phi(z)$. The primary motivations behind these requirements are that convexity facilitates computations, while calibration ensures the statistical consistency of the empirical estimator derived from $R_\phi$, as formally defined below.

**Definition 1.1** (Bartlett et al. (2006)). A loss function $\phi(\cdot)$ is classification-calibrated, if for every sequence of measurable function $f_n$ and every probability distribution on $\mathcal{X} \times \{\pm 1\}$,

$$R_\phi(f_n) \to \inf_f R_\phi(f) \text{ implies that } R(f_n) \to \inf_f R(f),$$

as $n$ approaches infinity.

According to Definition 1.1, a calibrated loss function $\phi$ guarantees that any sequence $f_n$ that optimizes $R_\phi$ will eventually also optimize $R$, thereby ensuring consistency in maximizing classification accuracy. To achieve this, the most commonly used and direct approach is ERM, which directly minimizes the empirical version of $R_\phi$ to obtain

[1]Department of Statistics, The Chinese University of Hong Kong. Correspondence to: Ben Dai <bendai@cuhk.edu.hk>.

*Proceedings of the 42nd International Conference on Machine Learning*, Vancouver, Canada. PMLR 267, 2025. Copyright 2025 by the author(s).

$f_n$. Specifically, given a training dataset $(\mathbf{x}_i, y_i)_{i=1}^n$, the $\phi$-classification framework is formulated as:

$$\widehat{f}_n = \underset{f \in \mathcal{F}}{\operatorname{argmin}} \ \widehat{R}_\phi(f), \quad \widehat{R}_\phi(f) := \frac{1}{n} \sum_{i=1}^n \phi(y_i f(\mathbf{x}_i)), \quad (2)$$

where $\mathcal{F} = \{f_{\boldsymbol{\theta}} : \boldsymbol{\theta} \in \boldsymbol{\Theta}\}$ is a candidate class of classification functions. For instance, $\mathcal{F}$ can be specified as, a Reproducing kernel Hilbert space (Aronszajn, 1950; Wahba, 2003), neural networks, or deep learning (DL) models (LeCun et al., 2015). Notably, most successful classification methods fall within the ERM framework of (2), utilizing various loss functions and functional spaces.

Given a functional space $\mathcal{F}$, the training process of ERM in (2) focus on optimizing the parameters $\boldsymbol{\theta}$ within $\boldsymbol{\Theta}$. Stochastic gradient descent (SGD; (Bottou, 1998; LeCun et al., 2002)) is widely adopted for its scalability and generalization when dealing with large-scale datasets and DL models. Specifically, in the $t$-th iteration, SGD randomly selects one or a batch of samples $(\mathbf{x}_{i_b}, y_{i_b})_{b=1}^B$ with the index set $\mathcal{I}_B$, and subsequently updates the model parameter $\boldsymbol{\theta}$ as:

$$\boldsymbol{\theta}^{(t+1)} = \boldsymbol{\theta}^{(t)} - \gamma \frac{1}{B} \sum_{i \in \mathcal{I}_B} \nabla_{\boldsymbol{\theta}} \phi(y_i f_{\boldsymbol{\theta}^{(t)}}(\mathbf{x}_i))$$

$$= \boldsymbol{\theta}^{(t)} - \gamma \frac{1}{B} \sum_{i \in \mathcal{I}_B} y_i \partial \phi(y_i f_{\boldsymbol{\theta}^{(t)}}(\mathbf{x}_i)) \nabla_{\boldsymbol{\theta}} f_{\boldsymbol{\theta}^{(t)}}(\mathbf{x}_i), \quad (3)$$

where $\gamma > 0$ represents a learning rate in SGD, the second equality follows from the chain rule, and $\nabla_{\boldsymbol{\theta}} f_{\boldsymbol{\theta}}(\mathbf{x})$ can be explicitly computed when the form of $f_{\boldsymbol{\theta}}$ or $\mathcal{F}$ is specified.

The ERM paradigm (2) with calibrated losses, when combined with ML models and optimized using SGD, has achieved tremendous success in numerous real-world applications. Notably, with deep neural networks, it has become a cornerstone of supervised classification in modern datasets (Goodfellow et al., 2016; Krizhevsky et al., 2012; He et al., 2016; Vaswani, 2017).

**Overfitting.** The overparameterized nature of deep neural networks is often necessary to capture complex patterns and various datasets in the real world, thereby achieving state-of-the-art performance. However, one of the most pervasive challenges in DL models is the problem of overfitting, where a model becomes overly specialized to the training data and struggles to generalize well to testing datasets. Particularly, DL models can even lead to a phenomenon where they perfectly fit the training data, achieving nearly zero training error, but typically, with a significant gap often persisting between the training (close to zero) and testing errors, a discrepancy attributable to overfitting. Given this fact, many regularization methods (c.f. Section 2) have been proposed, achieving remarkable improvements in alleviating overfitting in overparameterized models. The purpose of this

article is to also propose a novel regularization method ENSLOSS, which differs from existing regularization methods, or rather, regularizes the model from a different perspective.

**Our motivation.** The primary motivation for ENSLOSS stems from ensemble learning, but it specifically focuses the perspective of loss functions, applying the ensemble concept to combine various "valid" losses. As mentioned previously, numerous CC losses can act as a valid surrogate loss in (2), yielding favorable statistical properties in terms of the zero-one loss in (1). Yet, pinpointing the optimal surrogate loss in practical scenarios remains a challenge. A potentially effective idea is *loss ensembles*, by implementing an ensemble of classification functions fitted from various valid loss functions. However, for large models, particularly those involving deep learning, the computational cost associated with multiple training sessions can often be prohibitively expensive. It is worth mentioning that a similar computation challenge is also prevalent with model ensembles or model combination. This issue, has been effectively addressed by Dropout (Srivastava et al., 2014): by randomly taking different network structures during each SGD update, thereby achieving the outcome akin to model ensembles. In our content, we employ a loosely analogous of Dropout, adopt different surrogate losses in each SGD update to achieve the objective of loss ensembles. This motivating idea behind ENSLOSS is roughly outlined in Table 1.

## 2. Related Works

This section provides a literature review of related works on regularization methods for mitigating overfitting, as well as related ML literature focused on the loss function.

**Dropout.** One simple yet highly effective method for preventing overfitting is dropout (Srivastava et al., 2014). The key advantage of Dropout lies in its ability to simulate an ensemble approach during SGD updates, thereby mitigating overfitting without substantial computational costs. Thus, Dropout has become a standard component in many DL architectures, and its advantages have been widely recognized. Notably, the direction of ensemble in Dropout is achieved through different model architectures, whereas our method achieves ensemble through different "valid" loss functions. Thus, the proposed method and Dropout exhibit a complementary relationship and can be used simultaneously, as implemented in Section 4.4.

**Penalization methods.** Another approach to mitigate overfitting is to impose penalties or constraints (such as a $L_1/L_2$ norm) on model parameters, which aims to reduce model complexity and thus prevent overfitting (Hoerl & Kennard, 1970; Tibshirani, 1996; Zou & Hastie, 2005). Similar approaches include weight decay during SGD (Loshchilov & Hutter, 2017). The underlying intuition is to strike a balance

between model complexity and data fitting, thereby mitigating overfitting through the bias-variance tradeoff. Thus, these methods can also be seamlessly integrated with the proposed method, as demonstrated in Section 4.4.

**Classification-calibration.** Note that the zero-one loss (or accuracy) cannot be directly optimized due to its discontinuous nature, and thus, a surrogate loss is introduced to facilitate the computation. A natural question that arises is: how can we ensure that the classifier obtained under the new loss performs well in terms of accuracy? The answer leads to the definition of classification-calibration for a loss function (Definition 1.1). Meanwhile, a series of works (Lin, 2004; Zhang, 2004b; Lugosi & Vayatis, 2004; Bartlett et al., 2006) have finally summarized loss calibration to a simple if-and-only-if condition, as stated in Theorem 3.1. Calibration is an extensively validated condition through both empirical and theoretical consideration, and is widely regarded as a necessary minimal condition for a loss function.

**Post loss ensembles.** A straightforward approach to constructing loss ensembles is to fit separate classifiers for each calibrated losses (e.g., SVM and logistic regression) and then combining their outputs using simple ensemble methods, such as bagging, stacking, or voting (Breiman, 1996; Wolpert, 1992). This approach has empirically achieved satisfactory performance; however, its major drawback is that it requires refitting a classifier for each loss, resulting in substantial computational costs that make it impractical for large complex models.

**Loss Meta-learn.** A recently related topic is the learning of loss functions via the *meta-learn* framework in multiple-task learning or domain adaptation (Gonzalez & Miikkulainen, 2021; Bechtle et al., 2021; Gao et al., 2022; Raymond, 2024). These methods primarily employ a two-step approach: first, learning a loss function from source datasets/tasks via bilevel optimization under Model-Agnostic Meta-Learning (Finn et al., 2017), and then applying the learned loss to traditional ERM in the target tasks/datasets. While they share some similarities with our approach in relaxing the fixed loss in ERM, they are mainly applied in transfer learning and typically require additional source tasks or datasets to learn the loss, differing from our setting and objectives.

## 3. Calibrated Loss Ensembles

As previously mentioned in the introduction, the motivation behind ENSLOSS lies in incorporating different loss functions into SGD updates. The proposed method can be delineated roughly within an informal outline in Table 1 (refer to Algorithm 1 for the formal details).

**Key empirical results.** Before delving into the technical details of our method, we begin by providing a representative "epoch-vs-test_accuracy" curve (Figure 1), to underscore

Table 1. Stochastic calibrated loss ensembles under SGD. **Upper**: A standard SGD updates (based on a fixed surrogate loss). **Lower**: Informal outline for the proposed loss ensembles method (please refer to Algorithm 1 for the formal details).

| SGD + **Fixed Loss** |
| --- |
| For each iteration: |
| • batch sampling from a training set; 
 • implement SGD on batch samples and *a fixed surrogate loss*. |

| SGD + **Ensemble Loss** (ENSLOSS; our) |
| --- |
| For each iteration: |
| • batch sampling from a training set; 
 ✚ randomly generate a new "valid" surrogate loss; 
 ➜ implement SGD on batch samples and *the generated surrogate loss*. |

the notable advantages of our proposed method over fixed losses. The experimental results in Figure 1 and Section 4 are highly promising, suggesting that ENSLOSS has the potential to significantly improve the performance of the fixed loss framework, with this improvement exhibiting a universal nature across epochs and various network models. The detailed setup and comprehensive evaluation of our method's empirical performance, along with related exploratory experiments, can be found in Section 4.

Based on the empirical evidence of its promising performance, we are now prepared to discuss the proposed loss ensembles method in detail. Certainly, the generation of surrogate loss functions is not arbitrary; it must still satisfy the requirements for consistency or calibration (Lin, 2004; Zhang, 2004b; Bartlett et al., 2006). Furthermore, the impact of the loss function under SGD is solely reflected in the *loss-derivative*, as discussed in subsequent sections. Thus, our primary focus is on the development of conditions of "valid" losses or loss-derivatives, as well as how to randomly produce them.

### 3.1. Calibrated loss-derivative

In this section, we aim to list the conditions for a valid loss and transform them into loss-derivative conditions, thereby facilitating the usage in SGD training; ultimately, these conditions directly inspire the implementation of our algorithm, with the overall motivation illustrated in Figure 2.

As indicated in the literature (c.f. Section 2), the well-accepted conditions for a valid surrogate loss $\phi$ are: (i) convexity; (ii) calibration. The key observation of SGD in (3) is that the *impact of the loss function $\phi$ on SGD or other (sub)gradient-based algorithms is solely reflected in its loss-derivative $\partial\phi$*. Interestingly, the convexity and calibration

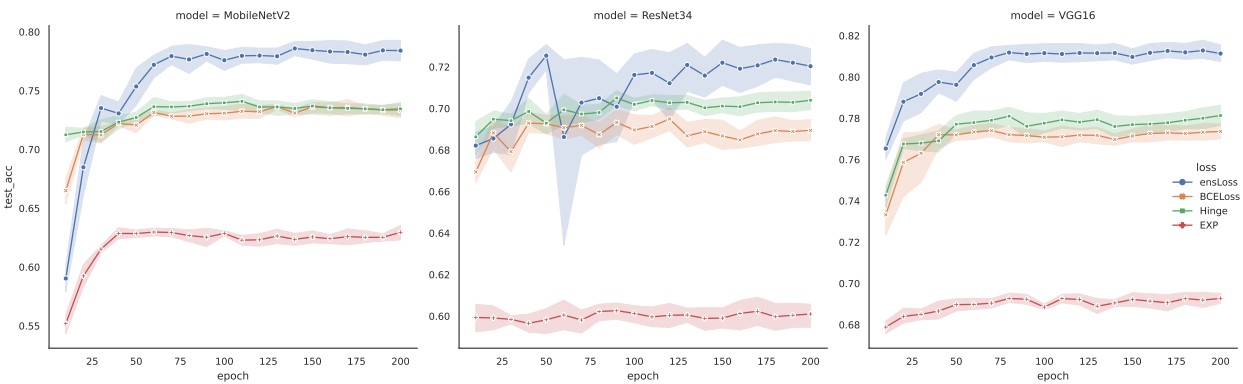

*Figure 1.* Comparison of epoch-vs-test_accuracy curves for various models on CIFAR2 (cat-dog) dataset using ENSLOSS (ours) and other fixed losses (logistic, hinge, and exponential losses). The training accuracy curves are omitted, as they have largely stabilized at 1 after few epochs. The pattern shown in the figure, where ENSLOSS consistently outperforms the fixed losses across epochs, is a phenomenon observed in almost all CIFAR10 label-pairs and the PCam dataset, and with different scales of ResNet, MobileNet, and VGG architectures.

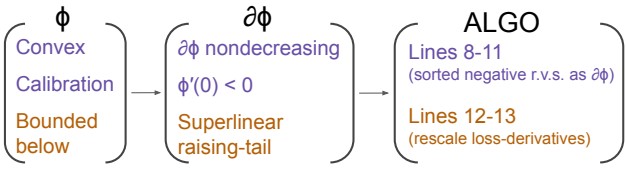

*Figure 2.* The overall motivation behind generating "valid" loss-derivatives in our algorithm: first transform the loss conditions (**left**) into loss-derivative (**middle**), thereby bypassing the loss and directly generating loss-derivatives in SGD algorithms (**right**).

conditions for $\phi$ can also be transformed to $\partial\phi$: (i) convexity can be ensured by stipulating that its loss-derivative is non-decreasing, and (ii) a series of literature (Lin, 2004; Zhang, 2004b; Lugosi & Vayatis, 2004; Bartlett et al., 2006) is finally summarized in the subsequent theorem, which provides a necessary and sufficient condition for calibration.

**Theorem 3.1** (Classification-calibration; (Bartlett et al., 2006)). *Let $\phi$ be convex. Then $\phi$ is classification-calibrated if and only if it is differentiable at 0 and $\phi'(0) < 0$.*

Theorem 3.1 effectively transfers the properties of convex calibration from $\phi$ to its loss-derivative $\partial\phi$, offering a straightforward and convenient approach to validate, design and implement a convex calibrated loss or loss-derivative under SGD implementation.

### 3.2. Superlinear raising-tail

Notably, there is one condition that could be easily overlooked yet remains crucial: the surrogate loss function $\phi$ must be nonnegative or *bounded below* (since we can always add a constant to make it a nonnegative loss, without affecting the optimization). Its importance lies in two-folds. Firstly, it directly influences calibration: the bounded below condition is a necessary condition of calibration, see

Corollary D.1. Secondly, although some unbounded below losses are proved calibrated in certain specific data distributions, yet they may introduce instability in the training process when using SGD in our numerical experiments, see more discussion in Appendix D. Therefore, we impose the bounded below condition for a loss $\phi$, or a form of regularity condition on the loss-gradient $\partial\phi$, see Lemma 3.2.

**Lemma 3.2.** *Let $\phi$ is convex and calibrated. If there exists a continuous function $g(z) > 0$ such that $\int_{z_0}^{\infty} g(z)dz$ converges, and $\partial\phi(z)/g(z)$ is nondecreasing when $z \geq z_0$ for some $z_0 > 0$. Then, $\phi$ is bounded below.*

Lemma 3.2 offers an implementation to translate the bounded below condition of loss functions into requirements on the loss-derivatives. Without loss of generality, we set $z_0 = 1$ in the subsequent discussion. This is analogous to the cut-off point in the hinge loss, which can be nullified through scaling $f(\mathbf{x})$ and does not significantly affect performance. Additionally, $g(z)$ can be chosen as $p$-integrals, i.e., $g(z) = 1/z^p$ for $p > 1$. Naturally, a smaller value of $p$ provides more flexibility to $\partial\phi$. Figure 3 illustrates the rights tails of loss-derivatives for some widely-used loss functions. Intuitively, Lemma 3.2 essentially indicates that the right tail of a valid loss-derivative needs to rise rapidly from $\phi'(0) < 0$ towards zero, either surpassing zero (as in the case of squared loss), or vanishing faster than $1/z$ when $z$ is large (ignoring the logarithm). We refer to this condition in Lemma 3.2 as a *superlinear raising-tailed* loss-derivative.

We now present all the conditions for the loss-derivative of a bounded below convex calibrated loss. Convexity implies that the loss-derivative is nondecreasing, while calibration requires the loss to be differentiable at 0 with $\phi'(0) < 0$. Additionally, the bounded below assumption yields that the loss-derivative exhibits a superlinear raising-tail. We refer this particular form of loss-derivatives as superlinear raising-tailed calibrated (RC) loss-derivatives. The follow-

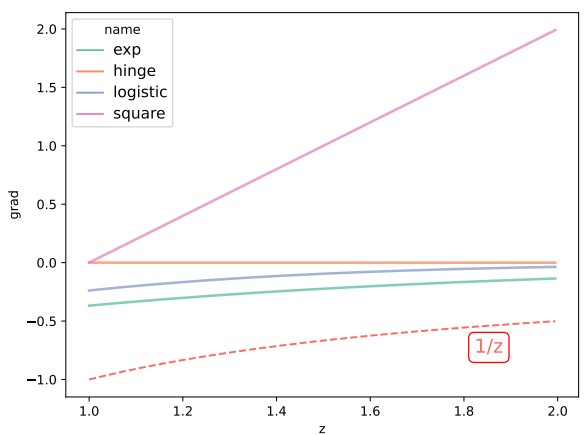

*Figure 3.* Plot of loss-derivatives of existing losses when $z > 1$.
**Conclusion.** Lemma 3.2 essentially indicates that the right tail of the loss-derivatives needs to rise rapidly from $\phi'(0) < 0$ towards zero, either surpassing zero (as in the case of squared loss) or vanishing faster than $1/z$ when $z$ is large (ignoring the logarithm).

ing lemma indicates that RC loss-derivatives essentially correspond to a bounded below convex calibrated loss.

**Lemma 3.3.** *Given a set of samples $(\mathbf{x}_i, y_i)_{i=1,\cdots,B}$ and a classification function $f$, let $z_i = y_i f(\mathbf{x}_i)$, and denote $\mathbf{g} = (g_1, \cdots, g_B)^\mathsf{T}$ as RC loss-derivatives, that is, satisfying the following conditions:*

1. *(Convexity) $g_i \leq g_j$ if $z_i < z_j$, and $g_i = g_j$ if $z_i = z_j$;*

2. *(Calibration) $g_i < 0$ if $z_i \leq 0$;*

3. *(Superlinear) $z_i^p g_i \leq z_j^p g_j$ if $1 \leq z_i < z_j$, for $p > 1$.*

*Then, there exists a bounded below convex calibrated loss function $\phi$, such that $\partial\phi(z_i) = g_i$ for all $i = 1, \cdots, B$.*

Lemma 3.3 sheds light upon the conditions for RC loss-derivatives (or its implicitly corresponding CC loss). Hence, it allows us to bypass the generation of loss and directly generate the loss-derivatives in SGD, thereby inspires *doubly stochastic gradients* in Algorithm 1.

Given that Conditions 1 and 3 require at least two samples to demonstrate the properties, our primarily focus on implementing our algorithm using mini-batch SGD. For sake of simplicity in implementation, we directly choose $p = 1$, as the numerical difference between $z$ and $z^p$ when $p$ is very close to 1 is exceedingly tiny. Our empirical experiments also demonstrate that $p = 1$ does not significantly affect the performance compared with $p$ close to 1, yet not performing superlinear raising tail adjustment on loss-derivatives can significantly impact the performance.

**Doubly stochastic gradients.** The most important implication of Lemma 3.3 is that it provides a guideline for generating RC loss-derivatives, as Conditions 1-3 are straightforward to satisfy. For example, we can obtain a set of RC

**Algorithm 1** (Minibatch) Calibrated ensemble SGD.

1: **Input:** a train set $\mathcal{D} = (x_i, y_i)_{i=1}^n$, a minibatch size $B$;
2: Initialize $\boldsymbol{\theta}$.
3: **for** number of epoches **do**
4:     /* **Minibatch sampling** */
5:     Sample a minibatch from $\mathcal{D}$ *without* replacement: $\mathcal{B} = \{(\mathbf{x}_{i_1}, y_{i_1}), \cdots, (\mathbf{x}_{i_B}, y_{i_B})\}$.
6:     Compute $\mathbf{z} = (z_1, \cdots, z_B)^\mathsf{T}$, where $z_b = y_{i_b} f_{\boldsymbol{\theta}}(\mathbf{x}_{i_b})$ for $b = 1, \cdots, B$.
7:     /* **Generate random RC loss-derivs** */
8:     /* calibration and convexity */
9:     Generate $\mathbf{g} = (g_1, \cdots, g_B)^\mathsf{T}$, where $g_b \overset{iid}{\sim} -\xi$, where $\xi$ is a *positive random variable* (accomplished through Algorithm 2)
10:     Sort $\mathbf{z}$ and $\mathbf{g}$ decreasingly, that is

$$z_{\pi(1)} > \cdots > z_{\pi(B)}, \quad g_{\sigma(1)} > \cdots > g_{\sigma(B)};$$

11:     (the derivative corresponding to $z_b$ is $g_{\sigma(\pi^{-1}(b))}$).
12:     /* bounded below */
13:     For $b = 1, \cdots, B$,

$$g_{\sigma(\pi^{-1}(b))} \leftarrow g_{\sigma(\pi^{-1}(b))} / z_b, \quad \text{if } z_b > 1.$$

14:     /* **Update parameters** */
15:     Compute gradients and update

$$\boldsymbol{\theta} \leftarrow \boldsymbol{\theta} - \frac{\gamma}{B} \sum_{b=1}^{B} y_{i_b} g_{\sigma(\pi^{-1}(b))} \nabla_{\boldsymbol{\theta}} f_{\boldsymbol{\theta}}(\mathbf{x}_{i_b})$$

16: **end for**
17: **Return** the estimated $\boldsymbol{\theta}$

loss-derivatives by sampling from a positive random variable $\xi$, then sorting and rescaling. Specifically, in Algorithm 1[1], the lines 8 to 11 (sampling and sorting) are dedicated to satisfy Conditions 1 and 2; the line 13 (rescaling) is for Condition 3. This implementation approach, which builds upon stochastic gradients by adding an another level of "stochasticity", is thus referred to as doubly stochastic gradients.

Note that Algorithm 1 does not explicitly implement Condition 2. In fact, for simplicity, we consider a sufficient condition that $g_i < 0$ for all $i = 1, \cdots, B$. This adaptation, made only for implementation simplicity, does not fundamentally alter the framework inspired by Lemma 3.3. Furthermore, the choice of the positive random variable $\xi$ impacts the diversity of the random loss-derivatives. To address this, we propose Algorithm 2 to generate distribution of $\xi$ using the inverse Box-Cox transformation (see discussion in Appendix A).

---

[1]We assume $z_i \neq z_j$, otherwise we can duplicate the derivative by merging and treating them as one sample.

### 3.3. Enhance loss diversity via the inverse Box-Cox transformation

In addition to considering the RC regularity and stochastic generating of loss-derivatives, it is also interesting to further boost the *diversity* of the loss functions or loss-derivatives generated from our proposed method (Algorithm 1). This could potentially improve the performance of loss ensembles, based on previous experience with ensemble learning (Breiman, 1996; 2001; Wood et al., 2023).

In fact, the diversity of loss functions in Algorithm 1 is partly associated with the loss-derivatives generated from the variety of positive random distributions. Consequently, a crucial consideration is to generate a sufficiently diverse range of positive random distributions. This naturally invokes the concept of the Box-Cox transformation (Box & Cox, 1964), which transforms (any) positive data via a power transformation with a hyperparameter $\lambda$ such that the transformed data closely approximates a normal distribution.

$$
\mathrm{BC}_\lambda(\xi) = \begin{cases} \lambda^{-1}(\xi^\lambda - 1), & \text{if } \lambda \neq 0, \\ \\ \log(\xi), & \text{if } \lambda = 0. \end{cases}
$$

Interestingly, the Box-Cox transformation is to transform a diverse range of positive random distributions into a normal distribution. This represents our exact "inverse" direction: we need to generate a sufficiently diverse range of positive random distributions. Consequently, we propose the *inverse Box-Cox transformation*, defined as $\mathrm{invBC}_\lambda(\cdot)$ in (4).

$$
\mathrm{invBC}_\lambda(\xi) = \begin{cases} (1 + \lambda\xi)_+^{1/\lambda}, & \text{if } \lambda \neq 0, \\ \\ \exp(\xi), & \text{if } \lambda = 0. \end{cases} \tag{4}
$$

On this ground, the final loss-derivatives can be generated in this manner, see Algorithm 2. First, we generate derivatives from a standard normal distribution, then using the inverse Box-Cox transformation (4) (with a random $\lambda$) transforms them into an arbitrary positive random distribution. This guarantees a variety in the loss function during the ensemble process over SGD. Here, the randomness of $\lambda$ is used to control a diverse range of loss-derivatives.

In our numerical experiments, we fix $\lambda = 0$ (no hyperparameter tuning) for a fair comparison, which corresponds to an exponential transformation of normally distributed random variables. In practice, to further enhance loss diversity, we can implement random sampling of $\lambda$ every $T$ epochs. This approach presents beneficial in certain cases, as demonstrated in our ablation studies presented in Table 6. However, determining a reasonable value of $T$ in a straightforward manner likely requires further investigation in future research.

---

**Algorithm 2** Inverse Box-Cox transform of loss-derivatives.

1: **Input:** a minibatch size $B$, a hyperparameter $\lambda$ (default $\lambda = 0$)
2: /* **Generate normal grad** */
3: Generate $\mathbf{g} = (g_1, \cdots, g_B)^\mathsf{T}$, where $g_b \overset{iid}{\sim} N(0, 1)$
4: /* **Inverse Box-Cox transformation** */

$$
g_b \leftarrow -\mathrm{invBC}_\lambda(g_b)
$$

5: **Return** the generated loss-derivatives $\mathbf{g} = (g_1, \cdots, g_B)^\mathsf{T}$.

---

### 3.4. Statistical consistency of loss ensembles

In this section, we establish a theoretical framework to analyze the statistical behavior and consistency of the proposed loss ensemble framework. Our idea comprises two primary steps: first, aligning the proposed method with a novel risk function; second, leveraging learning theory to evaluate the calibration and consistency of the risk function.

To proceed, we introduce relevant definitions to construct the corresponding risk function for the proposed method. Specifically, we denote $\mathcal{L}$ as a measurable space consisting of bounded below convex calibrated (BCC) losses. A random surrogate loss $\Phi$ is considered as a $\mathcal{L}$-valued random variable, where a loss $\phi$ represents an observation or sample of $\Phi$. Note that in our analysis, $\Phi$ is assumed independent of $(\mathbf{X}, Y)$; for detailed probabilistic definitions of random variables in functional spaces, refer to (Mourier, 1953; Vakhania et al., 2012). On this ground, we introduce the *calibrated ensemble risk function* as follows:

$$
\overline{R}(f) := \mathbb{E}_{\mathbf{X}, Y}\Big( \mathbb{E}_\Phi \Phi\big( Y f(\mathbf{X}) \big) \Big), \tag{5}
$$

where $\mathbb{E}_\Phi$ is the expectation taken with respect to $\Phi$. In this content, given a classification function $f_{\boldsymbol{\theta}}$, with a minibatch data $(\mathbf{x}_i, y_i)_{i \in \mathcal{I}_B}$ and the sampled loss $\Phi = \phi$, the stochastic gradient of $\overline{R}$ in SGD is defined as:

$$
\widehat{\mathbf{g}} = \frac{1}{|\mathcal{I}_B|} \sum_{i \in \mathcal{I}_B} \nabla_{\boldsymbol{\theta}} \phi\big( y_i f_{\boldsymbol{\theta}}(\mathbf{x}_i) \big). \tag{6}
$$

Thus, the proposed method (Algorithm 1) can be regarded as the SGD updating based on $\widehat{\mathbf{g}}$, and $\overline{R}(\cdot)$ is an appropriate risk function for characterizing the proposed method.

Next, we discuss the loss space $\mathcal{L}$. To ensure the calibration of the proposed method, we require that $\mathcal{L}$ is a measurable subspace of the collection of all BCC losses:

$$
\mathcal{L} \subset \big\{ \phi \text{ is convex} \mid \inf_z \phi(z) > -\infty, \; \phi'(0) < 0 \big\}.
$$

**Assumption 3.4.** Let $(\Omega, \mu)$ be a probability space and $\mathcal{L}$ be a measurable space. Suppose $\Phi : \Omega \to \mathcal{L}$ is a $\mathcal{L}$-valued random variable satisfies following conditions.

1. $\mathbb{E}\Phi(z)$ exists and $\mathbb{E}\Phi(z) < \infty$ for any given $z \in \mathbb{R}$.

2. There exists a random variable $U : \Omega \to \mathbb{R}$ such that $\Phi(z) \geq U$ a.s. for all $z$ and $\mathbb{E}U > -\infty$;

3. There exist a random variable $G : \Omega \to \mathbb{R}$ and a constant $\delta_0 > 0$, such that $|\partial\Phi(\delta)| \leq G$ a.s. for any $\delta \in [-\delta_0, \delta_0]$ and $\mathbb{E}G < \infty$.

Assumption 3.4 characterizes the feasibility of the loss space $\mathcal{L}$ and probability measure $\mu$, thereby establishing the scope of applicability for our proposed method. Notably, a finite loss space automatically satisfies Assumption 3.4.

**Lemma 3.5.** *If the loss space $\mathcal{L}$ is finite, then Assumption 3.4 is automatically satisfied.*

Indeed, $\mathcal{L}$ can be extended to a more general functional space, subject to the mild uniform assumptions in Assumption 3.4, which are necessary to preserve the completeness of BCC properties. For example, limiting over $\mathcal{L}$ without uniform assumptions may violate the calibration condition. In practice, during SGD training, the number of epochs is typically fixed, which implies that the number of ensemble losses implemented is also fixed. As a result, our analysis of the proposed method in practical scenarios can be exclusively focused on a finite $\mathcal{L}$ case.

Now, we have aligned the proposed method with the proposed ensemble risk (5). Hence, we can infer the statistical behavior of our method by analyzing $\overline{R}$. Of primary requirement is the calibration or Fisher-consistency, as formally stated in the following theorem.

**Theorem 3.6** (Calibration). *Suppose Assumption 3.4 holds, and $\overline{R}(\cdot)$ is defined as in (5) for any distributions $\mathbb{P}_{\mathbf{X},Y}$ and $\mathbb{P}_\Phi$, then for every sequence of measurable function $f_n$,*

$$\overline{R}(f_n) \to \inf_f \overline{R}(f) \quad \text{implies that} \quad R(f_n) \to \inf_f R(f). \tag{7}$$

*Moreover, the excess risk bound is provided as:*

$$R(f) - \inf_f R(f) \leq \psi^{-1}\big(\overline{R}(f) - \inf_f \overline{R}(f)\big), \tag{8}$$

*where $\psi^{-1}$ is the inverse function of $\psi$, and $\psi$ is defined as:*

$$\psi(\theta) = \mathbb{E}\big(\Phi(0)\big) - \inf_{\alpha \in \mathbb{R}} \mathbb{E}\big(\frac{1+\theta}{2}\Phi(\alpha) + \frac{1-\theta}{2}\Phi(-\alpha)\big).$$

Theorem 3.6 ensures the classification-calibration of the proposed method, indicating that minimizing the ensemble calibrated risk $\overline{R}$ would provide a reasonable surrogate for minimizing $R(f)$. Furthermore, the excess risk bound is also provided in (8), which enables presenting the relationship between $R(f) - R^*$ and $\overline{R}(f) - \overline{R}^*$ when the distribution of $\Phi$ is given.

In addition, there is a substantial amount of literature that discusses the convergence results based on (batch) SGD (Moulines & Bach, 2011; Shamir & Zhang, 2013; Fehrman et al., 2020; Garrigos & Gower, 2023). Given that the stochastic gradient $\widehat{\mathbf{g}}$ in (6) for the proposed method offers an unbiased estimate of $\nabla_{\boldsymbol{\theta}}\overline{R}(f_{\boldsymbol{\theta}})$, many existing SGD convergence results can be extended to our ensemble setting, ensuring that $\overline{R}(f_n) \to \inf_f \overline{R}(f)$.

By combining the convergence results from SGD and the calibration established in Theorem 3.6, we demonstrate that the proposed loss ensemble preserves statistical consistency for classification accuracy. We illustrate the usage of the theorem by a toy example in Example C.1 of Appendix C.

The proposed method appears to have an analogous effect to bagging (Bühlmann & Yu, 2002; Soloff et al., 2024). We next provide theoretical insights into a natural question: *what advantages do loss ensembles offer over a fixed loss approach?* We partially address this question by examining the *Rademacher complexity*. Specifically, given a classification function space $\mathcal{F}$, the Rademacher complexity of $\phi$-classification are defined as follows:

$$\mathrm{Rad}_\phi(\mathcal{F}) := \sup_{f \in \mathcal{F}} \Big| \frac{1}{n} \sum_{i=1}^n \tau_i \phi\big(Y_i f(\mathbf{X}_i)\big) \Big|,$$

where $(\tau_i)_{i=1}^n$ are i.i.d. Rademacher random variables independent of $(\mathbf{X}_i, Y_i)_{i=1}^n$. The Rademacher complexity plays a crucial role in many existing concentration inequalities (Talagrand, 1996a;b; Bousquet, 2002), determining the convergence rate of the excess risk (Bartlett & Mendelson, 2002) (with smaller values yielding a faster rate). On this ground, the corresponding Rademacher complexity for the proposed ensemble loss method can be formulated as:

$$\overline{\mathrm{Rad}}(\mathcal{F}) := \sup_{f \in \mathcal{F}} \Big| \frac{1}{n} \sum_{i=1}^n \tau_i \mathbb{E}_\Phi \Phi\big(Y_i f(\mathbf{X}_i)\big) \Big|$$
$$\leq \mathbb{E}_\Phi\big(\mathrm{Rad}_\Phi(\mathcal{F})\big), \tag{9}$$

where the inequality follows from the Jensen's inequality. This simple deduction yields a positive result, that is, the Rademacher complexity of the ensemble loss is no worse than the average based on the set of fixed losses. Yet, (9) only partially showcases potential benefits of loss ensemble, but it does not provide conclusive evidence of its superiority over fixed losses, as a comparison of their excess risk bounds is also crucial. In fact, ensemble loss appears suboptimal in terms of distribution-free excess risk bounds. Furthermore, achieving definitive and practical conclusions across specific datasets or distributions remains a longstanding challenge for statistical analysis. The development of more effective combining weights, as in ensemble learning (Yang, 2004; Audibert, 2004; Dalalyan & Tsybakov, 2007), may provide a promising solution for future research.

# 4. Experiments

This section presents experiments comparing ENSLOSS with fixed loss methods and assessing its compatibility with regularization methods. All Python codes is openly accessible at our GITHUB. All experimental results, up to the epoch level, are publicly available on our W&B projects ENSLOSS-IMG and ENSLOSS-TAB, enabling transparent and detailed tracking and analysis.

## 4.1. Datasets, models and losses

**Image datasets.** We present the empirical results for image benchmark datasets: the CIFAR10 (Krizhevsky et al., 2009) and the PatchCamelyon (PCam; (Veeling et al., 2018)). CIFAR10 was originally designed for multiclass image classification. In our study, we construct 45 binary CIFAR datasets, denoted as CIFAR2, by selecting all possible pairs of two classes from CIFAR10, which enables the evaluation of our method. PCam is a binary image classification dataset comprising 327,680 96x96 images from histopathologic scans of lymph node sections, each annotated with a label indicating the presence or absence of metastatic tissue. CIFAR and PCam datasets are widely recognized benchmarks in image classification, frequently employed in various studies, such as (He et al., 2016; Huang et al., 2017; Srinidhi et al., 2021).

**Tabular datasets.** We applied a filtering ($n \geq 1000, d \geq 1000$) across *all* OpenML (Vanschoren et al., 2014) binary classification dense datasets, resulting 14 datasets: Bioresponse, guillermo, riccardo, christine, hiva-agnostic, and 9 OVA datasets: Breast, Uterus, Ovary, Kidney, Lung, Omentum, Colon, Endometrium, and Prostate.

**Models.** To assess the effectiveness of the proposed method, we explore a range of commonly used neural network structures, including Multilayer Perceptrons (MLPs; (Hinton, 1990)) with varying depths for tabular data, as well as VGG (Simonyan & Zisserman, 2014), ResNet (He et al., 2016), and MobileNet (Sandler et al., 2018) for image data.

**Fixed losses.** ENSLOSS is benchmarked against with the ERM framework (2) using three widely adopted fixed classification losses: the logistic loss (BCE; binary cross entropy), the hinge loss (HINGE), and the exponential loss (EXP).

## 4.2. Evaluation

All experiments are replicated 5 (for image data) or 10 (for tabular data) times, and the resulting accuracy values are reported. Moreover, to evaluate the statistical significance, p-values are calculated using a one-tailed paired sample $t$-test, with the null and alternative hypotheses defined as:

$$H_0 : \text{Acc}_{\mathcal{A}} \leq \text{Acc}_{\mathcal{B}}, \quad H_1 : \text{Acc}_{\mathcal{A}} > \text{Acc}_{\mathcal{B}}, \quad (10)$$

where $\text{Acc}_{\mathcal{A}}$ and $\text{Acc}_{\mathcal{B}}$ are accuracies provided by two compared methods. A p-value of $\leq 0.05$ indicates strong evidence suggesting that $\mathcal{A}$ outperforms $\mathcal{B}$. Pairwise hypothesis tests are performed for each pair of methods. If a method exhibits statistical significance compared to all other methods, it will be highlighted in bold font in the tables. Our primary focus is on accuracy, but we also provide AUC results in our accompanying W&B projects and GitHub repository.

## 4.3. ENSLOSS vs fixed loss methods

This section presents our experimental results, wherein we examine the numerical performance of ENSLOSS and compare it with other fixed loss methods across various setups.

**Design.** The experiment design is straightforward: we compare various methods on 46 image datasets (including 45 CIFAR2 datasets and the PCam dataset) and 14 OpenML tabular datasets and using different network architectures. The implementation settings for each method are identical, with the only difference in loss functions.

**Results.** Due to space limitations, we only present the summary statistics of significant testing results across all datasets in Table 2 (for 14 tabular datasets) and Table 3 (for 45 CIFAR2 datasets). Moreover, some highlighting detailed empirical results are provided in Appendix B, and all performance metrics for all experiments are publicly accessible at the epoch level via our W&B projects.

**Conclusion.** The key findings are summarized as follows.

*Image data.* Tables 9 - 10 and Figure 4 demonstrate that ENSLOSS consistently outperforms existing fixed loss methods. (i) The improvement is universal across experiments. As shown in Table 9, ENSLOSS achieves *non-inferior* performance in all 45 CIFAR2 datasets, and significantly outperforms ALL other methods in at least 60% of the datasets. (ii) The improvement is remarkable, with substantial gains of 3.84% and 3.79% observed in CIFAR2 (cat-dog) and PCam, respectively, surpassing the best fixed loss method paired with the optimal network architecture.

*Tabular data.* Table 11 reveals that: (i) When dealing with overparameterized models, ENSLOSS tends to be a more desirable option compared to fixed losses, whereas for less complex models, ENSLOSS may underperform or outperform on certain datasets compared to the optimal fixed loss, yet it remains a viable alternative worth considering overall. (ii) The effectiveness of ENSLOSS exhibits a clear upward trend as model complexity increases, as evident from the performance comparison from MLP(1) to MLP(5).

The superiority of ENSLOSS is more pronounced in image data than in tabular data, likely attributable to the increased risk of overfitting associated in high-dimensional inputs and complex models characteristic of image datasets.

The improvement is also prominently reflected at the *epoch level*, particularly after the training accuracy for the pro-

posed ENSLOSS reaches or approaches one, as suggested by the epoch-vs-test_accuracy curves in Figure 1 (and those for all experiments in our W&B projects). This is crucial for practitioners: by specifying a relatively large number of epochs, ENSLOSS is a promising choice compared to fixed loss methods; furthermore, its training accuracy can sometimes serve as a key indicator for early-stopping, obviating the need of a validation set.

*Table 2.* The summary statistics of datasets exhibiting statistical significance when comparing the proposed ENSLOSS against all other fixed loss methods in 45 **CIFAR2** binary classification datasets (*provided by pairwise labels subset of CIFAR10*) are presented. The significance of "better", "no diff", and "worse" are suggested by the significance test, as described in Section 4.2.

| (ENSLOSS) MODELS | (vs BCE) | (vs EXP) | (vs HINGE) |
|---|---|---|---|
| | (better, no diff, worse) with $p < 0.05$ | | |
| ResNet34 | (41, 4, 0) | (45, 0, 0) | (36, 9, 0) |
| ResNet50 | (42, 3, 0) | (45, 0, 0) | (43, 2, 0) |
| ResNet101 | (39, 6, 0) | (45, 0, 0) | (40, 5, 0) |
| VGG16 | (36, 9, 0) | (45, 0, 0) | (29, 16, 0) |
| VGG19 | (36, 9, 0) | (45, 0, 0) | (27, 18, 0) |
| MobileNet | (45, 0, 0) | (45, 0, 0) | (44, 1, 0) |
| MobileNetV2 | (45, 0, 0) | (45, 0, 0) | (45, 0, 0) |

*Table 3.* The summary statistics of datasets exhibiting statistical significance when comparing the proposed ENSLOSS against all other fixed loss methods in 14 *OpenML* datasets are presented.

| (ENSLOSS) MODELS | (vs BCE) | (vs EXP) | (vs HINGE) |
|---|---|---|---|
| | (better, no diff, worse) with $p < 0.05$ | | |
| MLP(1) | (9, 4, 1) | (7, 5, 2) | (5, 4, 5) |
| MLP(3) | (7, 7, 0) | (8, 5, 1) | (9, 3, 2) |
| MLP(5) | (11, 3, 0) | (11, 2, 1) | (13, 0, 1) |

### 4.4. Compatibility of prevent-overfitting methods

As discussed in Section 2, the proposed ENSLOSS *complements* most existing prevent-overfitting methods, suggesting the potential for their simultaneous use. In this experiment, we empirically investigate the compatibility of ENSLOSS with widely used regularization methods: DROPOUT, L2-regularization (or equivalently weight decay, denoted as WEIGHTD), and data augmentation (DATAAUG; (Wong et al., 2016; Xu et al., 2016)).

**Design and results.** To illustrate the compatibility, we conduct on the CIFAR-2 (cat-dog) dataset and ResNet50, using the same experimental setup as the main experiment. Specifically, we compare the performance of ENSLOSS with fixed losses under various regularization methods, and the results are presented in Table 4, which illustrates their compatibility and effectiveness in preventing overfitting.

**Conclusion.** According to Table 4, our prior hypothesis is confirmed: ENSLOSS is compatible with other regularization methods, and their combination yields additional

*Table 4.* The averaged accuracy (with AUC included in our Github repository, exhibiting similar patterns) and their standard errors (in parentheses) for all methods with various **regularization** on the **CIFAR2 (cat-dog)** image dataset are presented. "HP" indicates the corresponding hyperparameter for each regularization method.

| REG | HP | BCE | EXP | HINGE | ENSLOSS |
|---|---|---|---|---|---|
| NO REG; baseline | — | 67.99(0.30) | 60.09(0.19) | 68.19(0.40) | 69.52(1.38) |
| WEIGHTD | 5e-5 | 67.64(0.14) | 60.43(0.23) | 68.26(0.65) | 71.01(1.04) |
| | 5e-4 | 67.59(0.35) | 61.57(0.56) | 67.57(0.28) | **72.04(0.35)** |
| | 5e-3 | 68.00(0.31) | 62.26(0.45) | 68.26(0.35) | 70.84(0.67) |
| DROPOUT | 0.1 | 67.50(0.39) | 60.70(0.34) | 67.89(0.30) | **72.48(0.22)** |
| | 0.2 | 68.13(0.54) | 60.02(0.52) | 67.78(0.44) | 70.08(1.28) |
| | 0.3 | 67.65(0.29) | 59.70(0.46) | 67.78(0.49) | 72.44(0.68) |
| DATAAUG | — | 79.22(0.12) | 58.96(0.31) | 80.47(0.26) | **83.00(0.25)** |

benefits in mitigating overfitting. Moreover, the advantages of ENSLOSS is further demonstrated by its consistent superior performance compared to other fixed losses, even with additional regularization methods. Another benefit of ENSLOSS is its relative insensitivity from time-consuming hyperparameter tuning, as a simple strategy of setting a large epoch often yields improved performance.

## 5. Conclusion

ENSLOSS is a framework designed to enhance ML performance by mitigating overfitting. The proposed method has shown potential to improve performance across a wide variety of datasets and models, particularly for overparameterized models. The primary motivation behind consists of two components: "ensemble" and the "calibration" of the loss functions. Therefore, this concept can be extensively applied to various ML problems, by identify the specific conditions for loss consistency or calibration. Fortunately, some consistency conditions have been extensively studied in the literature, including Section 4 in (Zhang, 2004a), Theorem 1 in (Zou et al., 2008), gamma-phi losses in (Wang & Scott, 2023), Theorem 7 in (Tewari & Bartlett, 2007) and encoding methods in (Lee et al., 2004) for multi-class classification, Theorem 2 in (Gao & Zhou, 2015) for bipartite ranking or AUC optimization, and Theorem 3.4 in (Scott, 2012) for asymmetric classification. In addition, new discussions regarding consistency, such as $H$-consistency (Awasthi et al., 2022), would also be intriguing in the context of ensembles with specific functional spaces.

A limitation of ENSLOSS is that it often requires more epochs to achieve stable training (see Table 8), resulting in longer training times. This issue is also seen in stochastic regularization methods like Dropout. Additionally, the selection of the positive random variable $\xi$ during random loss-derivative generation, currently done using the inverse Box-Cox transformation, needs further investigation.

## Acknowledgements

We thank the anonymous Area Chair and some reviewers for their valuable feedback, suggestions and support. This work was supported by the Hong Kong RGC-ECS Grant 24302422 and Hong Kong RGC Grant 14304823.

## Impact Statement

This paper presents work whose goal is to advance the field of Machine Learning. There are many potential societal consequences of our work, none of which we feel must be specifically highlighted here.

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

# A. Ablation studies

**The necessity of the bounded below condition.** The bounded below condition is an important yet easily overlooked condition. Here, we present ablation studies to demonstrate its significance. Our experimental setup is straightforward—we simply remove the superlinear raising-tail condition induced by the bounded below condition (Line 13 in Algorithm 1).

Table 5 presents typical training and test performance metrics at 20-epoch intervals for CIFAR2 (dog-cat) using VGG16, comparing models with and without this condition. Similar patterns are consistently observed across all datasets and network architectures.

| epoch | 20 | 40 | 60 | 80 | 100 | 120 | 140 | 160 | 180 | 200 |
|---|---|---|---|---|---|---|---|---|---|---|
| **w/o B-below** | | | | | | | | | | |
| Train Acc | 0.498 | 0.503 | 0.499 | 0.498 | 0.491 | 0.499 | 0.501 | 0.506 | 0.502 | 0.502 |
| Test Acc | 0.488 | 0.508 | 0.507 | 0.495 | 0.508 | 0.488 | 0.507 | 0.491 | 0.492 | 0.493 |
| **w/ B-below** | | | | | | | | | | |
| Train Acc | 0.961 | 1.000 | 1.000 | 1.000 | 1.000 | 1.000 | 1.000 | 1.000 | 1.000 | 1.000 |
| Test Acc | 0.741 | 0.826 | 0.827 | 0.827 | 0.827 | 0.827 | 0.829 | 0.827 | 0.826 | 0.828 |

*Table 5.* Comparison of training and testing accuracy with and without the bounded below (B-below) condition across different epochs. The results indicate that the absence of the bounded below condition leads to model fail to learn.

According to Table 5, without the bounded below condition, the model fails to learn, achieving only random-level performance ($\sim$50% accuracy) throughout training. In contrast, with the bounded below condition implemented, the model achieves high training accuracy (100%) and demonstrates good generalization with approximately 83% test accuracy.

From a loss-derivative perspective, the explanation is straightforward: Line 13 implements a superlinear loss-derivative that effectively discounts gradients for correctly classified samples ($z = yf(\mathbf{x}) > 1$). This focuses optimization efforts on misclassified or boundary samples, a common design principle in classification loss functions.

When this condition (Line 13) is reflected into a loss function as a boundedness below constraint, the result is somewhat counterintuitive and unexpected. However, within the framework of convexity and calibration, the boundedness below condition appears to take on additional significance: Lemma 3.2 suggests that this condition provides essential regularity for the right tail of the loss-derivative. Additionally, in the Appendix D, we provide a more comprehensive theoretical investigation of the importance of the bounded below condition for loss calibration.

**Randomly updating the inverse Box-Cox transformation.** The performance of ENSLOSS, based on ResNet50 and CIFAR2 datasets, with varying values of $T$, is presented in Table 6, under the same experimental setting as described in Section 4.3.

| DATASETS | fixed $\lambda = 0$ (used in Section 4) | $T = 10$ | $T = 20$ | $T = 50$ |
|---|---|---|---|---|
| CIFAR2 (cat-dog) | 70.04(1.21) | 70.87(0.72) | 71.48(0.62) | 70.22(1.11) |
| CIFAR2 (bird-cat) | 81.12(0.28) | 80.45(0.30) | 80.58(0.38) | 80.63(0.47) |
| CIFAR2 (cat-deer) | 83.11(0.29) | 83.05(0.12) | 82.82(0.25) | 83.47(0.15) |

*Table 6.* The averaged classification Accuracy and their standard errors (in parentheses), for different periods $T$ of randomly updating inverse Box-Cox transformation based on ResNet50 are reported for CIFAR2 datasets.

These preliminary experiments suggest that randomly updating $\lambda$ over epochs can be beneficial in certain cases; however, the effects are not particularly significant, and identifying the optimal value for this tuning parameter seems challenging. Consequently, we implement experiments in Section 4 with a fixed $\lambda = 0$, which corresponds to an exponential transformation of normally distributed random variables, and defer further investigation of this topic for future research.

**Baseline of post-ensemble over fixed losses.** We supplement our comparison by evaluating our approach against baselines of post-ensembling fixed losses. For illustration, we report test performance for model averaging (AVERAGE) and majority

| | BCE | EXP | HINGE | AVERAGE | VOTING | ENSLOSS |
|---|---|---|---|---|---|---|
| (Acc) | 67.53 (1.05) | 67.30 (0.38) | 67.90 (1.33) | 70.61 (1.09) | 69.67 (1.59) | **72.37 (1.04)** |
| (AUC) | 74.31 (0.73) | 73.94 (0.75) | 74.39 (0.93) | 77.63 (0.37) | — | **79.81 (1.14)** |

*Table 7.* The averaged classification Accuracy and AUC and their standard errors (in parentheses) of model combination methods in the image dataset **CIFAR2 (cat-dog)** based on ResNet50 are presented. Bold font is used to denote statistical significant improvements over ALL other competitors.

voting (VOTING) on CIFAR2 (dog-cat) using ResNet50 trained with three fixed losses. Similar patterns are observed across all datasets and network architectures. The results are presented in Table 7.

As shown, post-ensembling indeed improves single fixed loss methods, but the proposed ENSLOSS still performs better. The challenge with post-ensembling is that selecting too many fixed losses requires substantial computational costs of training, while combining too few losses produces less effect. A key advantage of ENSLOSS is that it requires only a single model training. Therefore, ENSLOSS differs from post-ensembling methods in practical application.

## B. Highlighting empirical results

In this section, we highlight some detailed empirical results of our proposed method, ENSLOSS, and compare it with other fixed loss methods on various image and tabular datasets. All detailed performance metrics for all experiments are publicly accessible at the epoch level via our Github repository and W&B projects.

**Training time comparison.** To more accurately compare the computational differences between ENSLOSS and fixed loss training, we provide detailed time comparisons between BCE, Hinge, and ENSLOSS in Table 8. Specifically, we report *the minimum number of epochs* required for training accuracy to stabilize (defined as remaining within an error margin of 0.005 thereafter) across different neural network architectures on the CIFAR2 (cat-dog) dataset.

| Loss | MobileNet | MobileNetV2 | ResNet101 | ResNet34 | ResNet50 | VGG16 | VGG19 |
|---|---|---|---|---|---|---|---|
| BCE | 90 | 80 | 80 | 70 | 45 | 15 | 45 |
| HINGE | 40 | 35 | 60 | 25 | 35 | 25 | 35 |
| ENSLOSS | 110 | 90 | 160 | 145 | 150 | 55 | 150 |

*Table 8.* Comparison of minimum epochs required for training accuracy to stabilize (within an error margin of 0.005) across different neural network architectures on the CIFAR2 (cat-dog) dataset.

As indicated in Table 8, ENSLOSS training typically requires 2-3 times more epochs than traditional BCE or Hinge loss training to stabilize the training procedure. This computational overhead is acceptable for practical applications, and we are actively exploring methods to reduce the training time in future work.

**Performance on CIFAR2 datasets.** Figure 4 illustrates the overall pattern of performance of all methods in 45 CIFAR2 binary classification datasets. Table 9 presents the performance results (accuracy and AUC values) for a representative single CIFAR2 dataset (CIFAR2 (cat-dog)). Additional detailed results for other network architectures and datasets available in our accompanying W&B projects and GitHub repository.

**Performance on PCam dataset.** Table 10 presents the performance results (accuracy and AUC values) for the PCam dataset for all different network architectures.

**Performance on OpenML datasets.** Table 11 presents the accuracy results for the 14 OpenML datasets using MLP(5), with additional detailed results for other network architectures available in our accompanying W&B projects and GitHub repository.

## C. Technical proofs

### C.1. Auxiliary definitions

To proceed, let us first introduce or recall the definitions and notations:

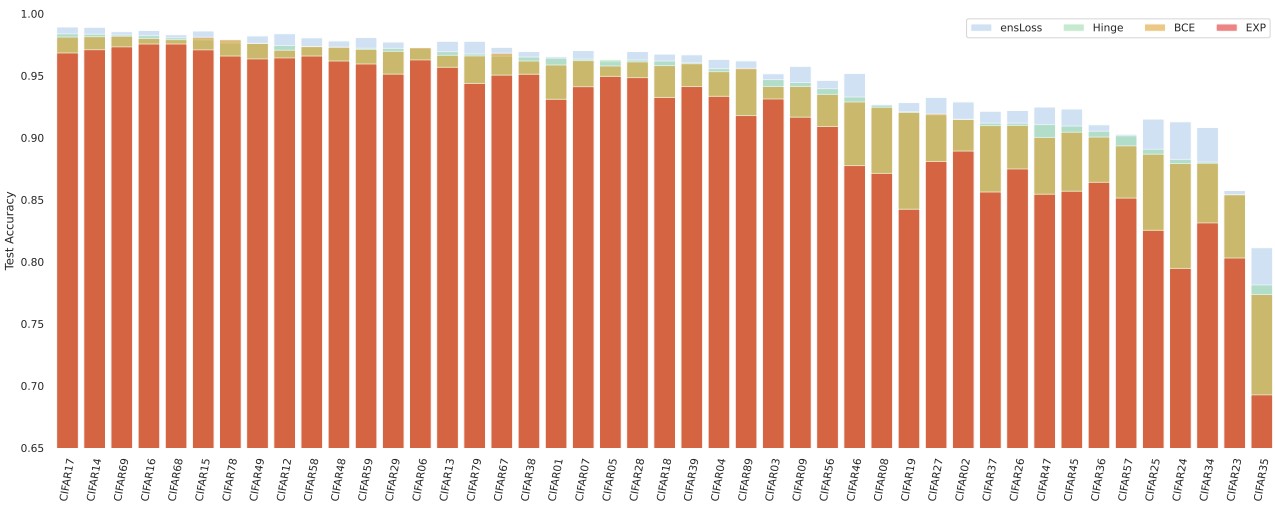

*Figure 4.* The overall pattern of performance (Accuracy) of ENSLOSS against all other fixed loss methods in 45 **CIFAR2** binary classification datasets (*provided by pairwise labels subset of CIFAR10*), based on VGG16, is illustrated. The $x$-axis represents label-paired binary CIFAR datasets, where, for example, CIFAR35 corresponds to the CIFAR2 (cat-dog) dataset.

- Classification probability: $\eta(\mathbf{X}) := \mathbb{P}(Y = 1|\mathbf{X})$.

- The Bayes classifier: $f^*(\mathbf{x}) = \mathrm{sgn}(\eta(\mathbf{x}) - 1/2)$.

- The pointwise minimization of $R$:

$$C_0(\eta, \alpha) = \eta \mathbf{1}(\alpha \geq 0) + (1 - \eta)\mathbf{1}(\alpha \leq 0), \quad H_0(\eta) = \inf_{\alpha \in \mathbb{R}} C_0(\eta, \alpha).$$

- The pointwise minimization of $R_\phi$:

$$C_\phi(\eta, \alpha) = \eta \phi(\alpha) + (1 - \eta)\phi(-\alpha), \quad H_\phi(\eta) = \inf_{\alpha \in \mathbb{R}} C_\phi(\eta, \alpha),$$

$$H_\phi^-(\eta) = \inf_{\alpha: \alpha(2\eta-1) \leq 0} C_\phi(\eta, \alpha), \quad H_\phi^+(\eta) = \inf_{\alpha: \alpha(2\eta-1) > 0} C_\phi(\eta, \alpha).$$

**Proof of Theorem 3.6**

*Proof.* We begin by defining the ensemble loss, denoted by $\bar{\phi}(z)$, as the expected value of $\Phi(z)$ with respect to the distribution of $\Phi$, i.e., $\bar{\phi}(z) := \mathbb{E}_\Phi \Phi(z)$. Note that the expectation is well-defined according to Assumption 3.4. Consequently, the calibrated ensemble risk function can be reformulated as follows:

$$\overline{R}(f) = \mathbb{E}_{\mathbf{X},Y}\big(\bar{\phi}(Yf(\mathbf{X}))\big).$$

Thus, we can rewrite the ensemble risk in the classical form of a fixed loss, namely, $\overline{R}(f) = R_{\bar{\phi}}(f)$, thereby enabling us to leverage Theorems 1 and 2 in (Bartlett et al., 2006) to facilitate statistical analysis of the proposed method. Therefore, it suffices to verify the BCC condition of the ensemble loss $\bar{\phi}$, that is, convexity, the bounded below condition, and having a negative derivative at 0.

To clarify, we sometimes denote $\Phi(z)$ as $\Phi(z, \omega)$ to emphasize that $\Phi : \Omega \to \mathcal{L}$ is a $\mathcal{L}$-valued random variable, equipped with a probability measure $\mu$. Now, we start with considering a simple finite space case, which provides insight into the underlying proof strategy.

**Specific case: a finite space.** Suppose $\mathcal{L} = \{\phi_1, \cdots, \phi_Q\}$ is a finite space, $\bar{\phi}(z)$ can be simplified as $\bar{\phi}(z) = \sum_{q=1}^{Q} \pi_q \phi_q(z)$, where $\pi_q = \mathbb{P}(\Phi = \phi_q) > 0$ and $\sum_{q=1}^{Q} \pi_q = 1$. Next, we check the BCC conditions of $\bar{\phi}$. (i) $\bar{\phi}$ is a convex combination of convex functions, thus $\bar{\phi}$ is convex; (ii) since $\phi_q$ is bounded below by $c_q$, thus $\bar{\phi}$ is bounded below by $\min_{q=1,\cdots,Q} c_q$; (iii) $\phi_q$ is differentiable at 0 for all $q$, thus $\bar{\phi}$ is also differentiable at 0, and $\bar{\phi}'(0) := \mathbb{E}\Phi'(0) = \sum_{q=1}^{Q} \pi_q \phi_q'(0) < 0$. Therefore, $\bar{\phi}$ satisfies the BCC conditions.

*Table 9.* The averaged classification Accuracy and AUC and their standard errors (in parentheses) of all methods in the image dataset **CIFAR2 (cat-dog)** are presented. Bold font is used to denote statistical significant improvements over ALL other competitors.

| MODELS | BCE | EXP | HINGE | ENSLOSS |
|---|---|---|---|---|
| **(Acc)** | | | | |
| ResNet34 | 68.94(0.45) | 60.10(0.44) | 70.39(0.37) | **72.03(0.71)** |
| ResNet50 | 67.59(0.35) | 61.57(0.56) | 67.57(0.28) | **72.04(0.35)** |
| ResNet101 | 67.32(0.23) | 53.57(0.41) | 67.12(0.32) | **70.07(0.90)** |
| VGG16 | 77.36(0.68) | 69.27(0.44) | 78.13(0.87) | **81.13(0.77)** |
| VGG19 | 76.96(1.03) | 66.38(1.17) | 78.06(0.68) | **80.57(0.76)** |
| MobileNet | 66.77(0.86) | 55.89(1.69) | 67.66(1.03) | **69.98(1.08)** |
| MobileNetV2 | 73.34(1.12) | 62.94(1.09) | 73.45(1.02) | **78.40(1.56)** |
| **(AUC)** | | | | |
| ResNet34 | 75.97(0.44) | 64.39(0.52) | 76.02(0.32) | **79.24(1.09)** |
| ResNet50 | 73.96(0.29) | 65.52(0.60) | 74.37(0.23) | **79.40(0.31)** |
| ResNet101 | 73.35(0.24) | 54.88(0.34) | 68.61(0.66) | **76.60(0.87)** |
| VGG16 | 85.42(0.71) | 76.13(0.94) | 86.20(0.63) | **89.54(0.29)** |
| VGG19 | 85.61(0.97) | 72.88(1.12) | 84.47(1.07) | **87.55(1.08)** |
| MobileNet | 73.12(0.97) | 58.51(1.77) | 73.96(1.34) | **76.61(1.23)** |
| MobileNetV2 | 81.29(0.96) | 67.53(1.11) | 81.31(0.74) | **86.22(1.37)** |

*Table 10.* The averaged classification Accuracy and AUC and their standard errors (in parentheses) of all methods in the image dataset **PCam** are presented. Bold font is used to denote statistical significant improvements over ALL other competitors.

| MODELS | BCE | EXP | HINGE | ENSLOSS |
|---|---|---|---|---|
| **(Acc)** | | | | |
| ResNet34 | 76.91(0.52) | 73.78(0.52) | 77.20(0.18) | **82.33(0.30)** |
| ResNet50 | 77.23(0.51) | 74.10(0.49) | 77.96(0.34) | **82.00(0.07)** |
| VGG16 | 80.97(0.25) | 77.11(0.50) | 82.69(0.30) | **85.77(0.35)** |
| VGG19 | 81.58(0.25) | 76.13(0.35) | 82.77(0.41) | **85.91(0.19)** |
| **(AUC)** | | | | |
| ResNet34 | 88.69(0.34) | 83.30(0.57) | 76.11(0.37) | **92.24(0.13)** |
| ResNet50 | 88.75(0.30) | 83.51(0.46) | 77.24(0.67) | **92.07(0.49)** |
| VGG16 | 93.35(0.26) | 88.77(0.59) | 86.18(0.56) | **95.44(0.24)** |
| VGG19 | 93.49(0.17) | 87.89(0.46) | 84.09(0.60) | **95.51(0.14)** |

**General cases.** Now, suppose $\mathcal{L}$ is a general measurable BCC subspace satisfying Assumption 3.4. The crucial issue is whether the BCC conditions are complete in the space $\mathcal{L}$ over limiting. Let us check the BCC conditions of $\bar{\phi}$. (i) Convexity. Note that

$$\bar{\phi}\big(\lambda z_1 + (1-\lambda)z_2\big) = \int_\Omega \Phi\big(\lambda z_1 + (1-\lambda)z_2, \omega\big) d\mu(\omega) \leq \int_\Omega \lambda\Phi(z_1, \omega) + (1-\lambda)\Phi(z_2, \omega) d\mu(\omega)$$
$$= \lambda\bar{\phi}(z_1) + (1-\lambda)\bar{\phi}(z_2),$$

where the inequality follows from the fact that $\mathcal{L}$ is a BCC subspace and thus $\Phi(z, \omega)$ is convex for any $\omega \in \Omega$. Thus, $\bar{\phi}$ is a convex function. (ii) Bounded below.

$$\bar{\phi}(z) = \int_\Omega \Phi(z, \omega) d\mu(\omega) \geq \int_\Omega U(\omega) d\mu(\omega) = \mathbb{E}U > -\infty,$$

where the inequality follow from the second condition in Assumption 3.4 such that $\Phi(z, \omega) \geq U(\omega)$ for any $z$ almost surely.

(iii) Calibration. For every sequence $\delta_n$ with $\delta_n \to 0$, without loss of generality, we assume $|\delta_n| \leq \delta_0$, which can be satisfied for sufficiently large $n$. We now check the definition of differentiability of $\bar{\phi}$ at $z = 0$:

$$\lim_{n\to\infty} \frac{\bar{\phi}(\delta_n) - \bar{\phi}(0)}{\delta_n} = \lim_{n\to\infty} \int_\Omega \frac{\Phi(\delta_n, \omega) - \Phi(0, \omega)}{\delta_n} d\mu(\omega) = \lim_{n\to\infty} \int_\Omega G_n(\omega) d\mu(\omega),$$

*Table 11.* The averaged classification Accuracy and its standard errors (in parentheses) of all methods with MLP(5) in 14 *OpenML* datasets are presented. Bold font is used to denote statistical significant improvements over ALL other competitors.

| **MLP**(5) | $(n, d) \times 10^3$ | BCE | EXP | HINGE | ENSLOSS (our) |
|---|---|---|---|---|---|
| Bioresponse | (3.75, 1.78) | 76.84(1.33) | 77.49(1.44) | 76.03(0.67) | 77.18(1.18) |
| guillermo | (20.0, 4.30) | 70.35(0.44) | 70.26(0.67) | 69.67(0.63) | **75.34(0.78)** |
| riccardo | (20.0, 4.30) | 98.68(0.21) | 98.69(0.13) | 98.62(0.23) | **99.14(0.23)** |
| hiva-agnostic | (4.23, 1.62) | 91.02(0.85) | 91.65(1.30) | **95.55(0.53)** | 90.61(1.49) |
| christine | (5.42, 1.64) | 69.62(1.07) | 69.42(1.30) | 67.48(0.72) | 69.94(0.93) |
| OVA-Breast | (1.54, 10.9) | 94.27(1.33) | 94.38(1.41) | 92.61(1.75) | **95.45(1.30)** |
| OVA-Uterus | (1.54, 10.9) | 80.54(1.54) | 82.09(1.50) | 84.22(1.75) | **86.68(1.66)** |
| OVA-Ovary | (1.54, 10.9) | 81.83(1.69) | 82.82(2.19) | 82.76(1.69) | **87.16(1.40)** |
| OVA-Kidney | (1.54, 10.9) | 97.59(0.83) | 97.72(0.65) | 96.47(0.95) | **98.06(0.48)** |
| OVA-Lung | (1.54, 10.9) | 88.17(1.70) | 89.31(2.36) | 89.76(1.53) | **93.00(1.31)** |
| OVA-Om | (1.54, 10.9) | 71.42(3.53) | 74.91(1.84) | 79.25(2.19) | **82.00(1.98)** |
| OVA-Colon | (1.54, 10.9) | 95.73(0.87) | 95.73(0.88) | 95.15(0.79) | **96.27(0.63)** |
| OVA-En | (1.54, 10.9) | 71.66(3.79) | 74.33(1.73) | 81.68(1.87) | **83.19(2.01)** |
| OVA-Prostate | (1.54, 10.9) | 97.39(0.51) | 96.96(0.77) | 97.22(0.84) | **97.93(0.60)** |

where $G_n(\omega) := \big(\Phi(\delta_n, \omega) - \Phi(0, \omega)\big)/\delta_n$. Next, the proof involves usage of the Dominated Convergence Theorem (c.f. Theorem 1.19 in (Evans, 2018)) to exchange the limit and the integration. To proceed, since $\Phi(z, \omega)$ is differentiable at 0 for any $\omega \in \Omega$, it follows that $\lim_{n \to \infty} G_n(\omega) = \Phi'(0, \omega)$ pointwise. Note that $\Phi(z, \omega)$ is convex w.r.t. $z$ for any $\omega \in \Omega$, then by the definition of the subderivative, we have,

$$\Phi(\delta_n, \omega) - \Phi(0, \omega) \geq \Phi'(0, \omega)\delta_n, \text{ and } \Phi(0, \omega) - \Phi(\delta_n, \omega) \geq -\partial\Phi(\delta_n, \omega)\delta_n,$$

which yields that

$$|G_n(\omega)| \leq \max\big(|\partial\Phi(\delta_n, \omega)|, |\Phi'(0, \omega)|\big) \leq \sup_{|\delta| \leq \delta_0} |\partial\Phi(\delta, \omega)| \leq G(\omega),$$

$$\int_\Omega G(\omega)d\mu(\omega) = \mathbb{E}G < \infty.$$

Thus, $G_n$ is dominated by $G$, then the Dominated Convergence Theorem yields that the limit exists and equals to,

$$\lim_{n \to \infty} \frac{\overline{\phi}(\delta_n) - \overline{\phi}(0)}{\delta_n} = \int_\Omega \lim_{n \to \infty} G_n(\omega)d\mu(\omega) = \int_\Omega \Phi'(0, \omega)d\mu(\omega) = \mathbb{E}\big(\Phi'(0)\big) := \overline{\phi}'(0) < 0,$$

where the last inequality follows from that $\Phi'(0, \omega) < 0$ for all $\omega \in \Omega$.

In summary, we have proved that $\overline{\phi} : \mathbb{R} \to \mathbb{R}$ satisfies the BCC conditions. Consequently, according to Theorem 1 and Theorem 2 (part 2) in (Bartlett et al., 2006), we can draw the following conclusions:

1. $\overline{\phi}$ is classification-calibrated, as stated in Definition 1.1.

2. The excess risk bound of $\overline{R}(\cdot)$ is provided as:

$$R(f) - R^* \leq \psi^{-1}\big(\overline{R}(f) - \overline{R}^*\big),$$

   where $\psi$ is defined as:

$$\psi(\theta) = \overline{\phi}(0) - H_{\overline{\phi}}\big(\frac{1 + \theta}{2}\big) = \mathbb{E}\big(\Phi(0)\big) - \overline{H}\big(\frac{1 + \theta}{2}\big),$$

   where $\overline{H}(\eta) := \inf_{\alpha \in \mathbb{R}} \mathbb{E}_\Phi C_\Phi(\eta, \alpha)$.

This completes the proof. □

**Proof of Lemma 3.2**

*Proof.* Since $\phi$ is convex and calibrated, $\phi'(0) < 0$, and $\phi(z) \geq \phi(0) + \phi'(0)z \geq \phi(0)$ when $z \leq 0$, thus $\phi(z)$ is bounded below when $z \leq 0$. Next, we aim to establish a lower bound of $\phi(z)$ for $z > 0$.

**CASE 1.** If there exists a constant $z_0 > 0$ such that $\partial\phi(z_0) \geq 0$, then $\phi(z) \geq \phi(z_0) + \partial\phi(z_0)(z - z_0) \geq \phi(z_0)$ for $z \geq z_0$. For $0 \leq z \leq z_0$, $\phi(z)$ is bounded according to the boundedness theorem. Then, $\phi(z)$ is bounded below.

**CASE 2.** If $\partial\phi(z) < 0$ for all $z > 0$. Then, for any $z > 0$, we partition $[0, z]$ as $n$ intervals with $d_0 = 0, d_1 = z/n, \cdots, d_{n-1} = (n-1)z/n, d_n = z$, for the $i$-th interval, we have

$$\phi(d_{i+1}) \geq \phi(d_i) + \partial\phi(d_i)z/n.$$

Taking the summation for both sides, and $\partial\phi(z)$ is nondecreasing according to Lemma C.2,

$$\phi(z) = \phi(d_n) \geq \phi(d_0) + \sum_{i=0}^{n-1} \frac{\partial\phi(d_i)z}{n} = \phi(0) + \frac{\phi'(0)z}{n} + \sum_{i=1}^{n-1} \frac{\partial\phi(d_i)z}{n}$$

$$\geq \phi(0) + \frac{\phi'(0)z}{n} + \int_0^z \partial\phi(z)dz \geq \phi(0) + \frac{\phi'(0)z}{n} + \int_0^\infty \partial\phi(z)dz,$$

where the second last inequality follows from the integral test, and the last inequality again follows from Lemma C.2. Taking the limit for $n \to \infty$, for any $z > 0$, we have

$$\phi(z) = \lim_{n\to\infty} \phi(z) \geq \phi(0) + \int_0^\infty \partial\phi(z)dz = \phi(0) + \int_0^{z_0} \partial\phi(z)dz + \int_{z_0}^\infty \frac{\partial\phi(z)}{g(z)}g(z)dz$$

$$\geq \phi(0) + \phi'(0)z_0 + \frac{\partial\phi(z_0)}{g(z_0)}\int_{z_0}^\infty g(z)dz \geq \phi(0) + \phi'(0)z_0 + \frac{\phi'(0)}{g(z_0)}\int_{z_0}^\infty g(z)dz > -\infty.$$

The desirable result then follows. □

**Proof of Lemma 3.3**

*Proof.* Before proceed, we add $z = 0$ into the batch points as $\{z_1, \cdots, z_{B+1}\}$. Without loss generality, we assume (i) $z_i \neq z_j$, as we can duplicate the gradient by merging and treating them as one point; and assume that (ii) $z_1 < z_2 \cdots < z_{B+1}$ and $z_{b_0} = 0$, as we can always sort the batch points. Next, we also design and add the loss-derivative for $z_{b_0} = 0$ into the given loss-derivatives $\mathbf{g}$ to form a new gradient vector $\tilde{\mathbf{g}}$, specifically,

$$\tilde{g}_i := g_i, \quad \text{if } i < z_0, \qquad \tilde{g}_i := g_{i-1} \quad \text{if } i > z_0,$$

where $\tilde{g}_{b_0} = \min(\tilde{g}_{b_0-1}/2, (\tilde{g}_{b_0-1} + \tilde{g}_{b_0+1})/2)$, if $b_0 > 1$; $\tilde{g}_{b_0} = \min(\tilde{g}_{b_0+1}, -1)$, if $b_0 = 1$. Hence, $\tilde{g}_{b_0} < 0$ and $\tilde{g}_{b_0-1} \leq \tilde{g}_{b_0} \leq \tilde{g}_{b_0+1}$.

Then, we define $u_1 = (z_1 + z_2)/2, \cdots, u_i = (z_i + z_{i+1})/2, \cdots, u_B = (z_B + z_{B+1})/2, u_{B+1} = z_{B+1} + 1$, and the corresponding loss function $\phi$ can be formulated as follows.

$$\phi(z) : \begin{cases} l_1(z) = \tilde{g}_1 z, & \text{if } z \leq u_1, \\ l_2(z) = \tilde{g}_2(z - u_1) + l_1(u_1), & \text{if } u_1 < z \leq u_2, \\ \cdots, \\ l_{b_0}(z) = \tilde{g}_{b_0}(z - u_{b_0-1}) + l_{b_0-1}(u_{b_0-1}), & \text{if } u_{b_0-1} < z \leq u_{b_0}, \\ \cdots, \\ l_{B+1}(z) = \tilde{g}_{B+1}(z - u_B) + l_B(u_B), & \text{if } u_B < z \leq u_{B+1}, \\ l_{B+2}(z) = \max(\tilde{g}_{B+1}, 1)(z - u_{B+1}) + l_{B+1}(u_{B+1}), & \text{if } z > u_{B+1}. \end{cases}$$

By definition, $\phi$ provides the loss-derivatives for original sample points, that is, $\phi'(z_i) = g_i$ for all $i \neq b_0$. Now, we verify that $\phi(z)$ is a BCC loss. Specifically, (i) $\phi(z)$ is a continuous piecewise linear function with coefficients $\tilde{g}_1 \leq \tilde{g}_2 \leq \cdots \leq \tilde{g}_{B+1} \leq \max(\tilde{g}_{B+1}, 1)$ according to Condition 1 and the definition of $\tilde{\mathbf{g}}$. $\phi(z)$ is a convex function. (ii) Note that $z_{b_0} = 0 \in (u_{b_0-1}, u_{b_0}]$, hence $\phi(z)$ is differentiable at 0 and $\phi'(0) = l'_{b_0}(0) = \tilde{g}_{b_0} < 0$. $\phi$ is calibrated. (iii) $\phi(z)$ is increasing function when $z \geq u_{B+1}$, thus it is bounded below. This completes the proof. □

**Proof of Lemma 3.5**

*Proof.* Suppose $\mathcal{L} = \{\phi_1, \cdots, \phi_Q\}$ is a finite space, then $\mathbb{E}\Phi(z)$ is defined as $\mathbb{E}\Phi(z) := \sum_{q=1}^{Q} \pi_q \phi_q(z) < \infty$, where $\pi_q = \mathbb{P}(\Phi = \phi_q) > 0$ and $\sum_{q=1}^{Q} \pi_q = 1$, which leads to Condition 1. Since $\phi_q$ is bounded below, that is, there exists a constant $U_q > -\infty$, such that $\phi_q(z) \geq U_q$, thus $\mathbb{E}\Phi(z) \geq \min_{q=1,\cdots,Q}(U_q) =: U > -\infty$, which leads to Condition 2. Finally, $\phi_q$ is convex and differentiable at 0, then $\partial\phi_q(z)$ converges to $\phi_q'(0)$ when $z \to 0$, see Corollary 4.2.3 in (Hiriart-Urruty & Lemaréchal, 1996). Therefore, for each $q = 1, \cdots, Q$, there exists constants $\delta_q > 0$ and $G_q > 0$, such that $|\partial\phi_q(z)| \leq G_q$ for all $z \in [-\delta_q, \delta_q]$, which leads to Condition 3 by the fact that $|\partial\phi_q(z)| \leq \max_{q=1,\cdots,Q} G_q =: G < \infty$ for $z \in [-\delta_0, \delta_0]$ with $\delta_0 = \min_{q=1,\cdots,Q} \delta_q$ for all $q = 1, \cdots, Q$. This completes the proof. $\qquad\square$

**Auxiliary lemmas**

**Lemma C.1.** *For $C_\phi(\eta, \alpha)$, $H_\phi(\eta)$, $H_\phi^-(\eta)$ and $H_\phi^+(\eta)$ defined in Appendix C.1 with a convex and bounded below $\phi$, then*

    *a. $C_\phi(\eta, \alpha)$ is convex with respect to $\alpha$.*

    *b. For any $\eta \neq 1/2$, $H_\phi^-(\eta) > H_\phi(\eta)$ is equivalent to $H_\phi^-(\eta) > H_\phi^+(\eta)$.*

    *c. If $\partial\phi(0) < 0$, then $H_\phi^-(\eta) = \phi(0)$.*

*Proof.* (a). Since $0 \leq \eta \leq 1$, $C_\phi(\eta, \alpha) = \eta\phi(\alpha) + (1-\eta)\phi(-\alpha)$ is a convex combination of $\phi(\alpha)$ and $\phi(-\alpha)$. Then, $C_\phi(\eta, \alpha)$ is convex with respect to $\alpha$ since both $\phi(\alpha)$ and $\phi(-\alpha)$ are both convex.

(b). Denote $\mathcal{A} = \{\alpha | \alpha(2\eta - 1) \leq 0\}$, then $H_\phi(\eta) = \inf_{\alpha \in \mathcal{A} \cup \mathcal{A}^c} C_\phi(\eta, \alpha)$, $H_\phi^-(\eta) = \inf_{\alpha \in \mathcal{A}} C_\phi(\eta, \alpha)$, and $H_\phi^+(\eta) = \inf_{\alpha \in \mathcal{A}^c} C_\phi(\eta, \alpha)$. Given that $\eta \neq 1/2$, both $\mathcal{A}$ and $\mathcal{A}^c$ are nonempty.

We first show that $H_\phi(\eta) = \min(H_\phi^-(\eta), H_\phi^+(\eta))$. Note that $H_\phi(\eta) \leq H_\phi^-(\eta)$ and $H_\phi(\eta) \leq H_\phi^+(\eta)$, then $H_\phi(\eta) \leq \min(H_\phi^-(\eta), H_\phi^+(\eta))$. On the other hand, for any $\alpha \in \mathbb{R}$, then either $\alpha \in \mathcal{A}$ or $\alpha \in \mathcal{A}^c$, thus $C_\phi(\eta, \alpha) \geq \min(H_\phi^-(\eta), H_\phi^+(\eta))$, and $H_\phi(\eta) \geq \min(H_\phi^-(\eta), H_\phi^+(\eta))$.

Now, $H_\phi^-(\eta) > H_\phi(\eta) = \min(H_\phi^-(\eta), H_\phi^+(\eta))$ if and only if $H_\phi^+(\eta) < H_\phi^-(\eta)$.

(c). Since $\phi$ is convex, $C_\phi(\eta, \alpha) = \eta\phi(\alpha) + (1-\eta)\phi(-\alpha) \geq \phi\big((2\eta-1)\alpha\big)$. Hence,

$$\phi(0) = C_\phi(\eta, 0) \geq H_\phi^-(\eta) = \inf_{\alpha(2\eta-1) \leq 0} C_\phi(\eta, \alpha) \geq \inf_{z \leq 0} \phi(z) = \phi(0),$$

where the last equality follows from the fact that $\phi$ is convex and $\phi(z) \geq \phi(0) + \partial\phi(0)z \geq \phi(0)$ when $z \leq 0$. The desirable result then follows. $\qquad\square$

**Lemma C.2.** *Let $\phi : \mathbb{R} \to \mathbb{R}$ be a convex function. Then*

    *a. The sub-derivative $\partial\phi$ is nondecreasing.*

    *b. If $\phi(x) > \phi(y)$ for some $(x, y)$, then for any $z$ between $x$ and $y$ (excluding $x$ and $y$), $\phi(x) > \phi(z)$.*

*Proof.* (a). By the definition of sub-derivative, for any $y > x$, we have

$$\phi(y) \geq \phi(x) + \phi'(x)(y - x), \quad \phi(x) \geq \phi(y) + \phi'(y)(x - y),$$

providing that $(y - x)(\phi'(x) - \phi'(y)) \leq 0$. Thus, $\phi'$ is a nondecreasing function.

(b). For any $z$ between $x$ and $y$, there exists $\lambda > 0$, such that $z = \lambda x + (1 - \lambda)y$, then

$$\phi(z) = \phi(\lambda x + (1 - \lambda)y) \leq \lambda\phi(x) + (1 - \lambda)\phi(y) < \phi(x).$$

$\qquad\square$

**An example illustrating Theorem 3.6**

Let $\mathcal{L} = \{\phi_1, \phi_2\}$, where $\phi_1(z) = \exp(-z)$ is the exponential loss, and $\phi_2(z) = \log(1 + e^{-2z})$ is the logistic loss, and $\mathbb{P}(\Phi = \phi_q) = \pi_q > 0$, then $\overline{R}(\cdot)$ is classification-calibrated, and the excess risk bound is provided as:

$$\psi\big(R(f) - R^*\big) \leq \overline{R}(f) - \overline{R}^*,$$

where $\psi(\theta) = \pi_1\big(1 - \sqrt{1 - \theta^2}\big) + \pi_2/2\big((1 - \theta)\log(1 - \theta) + (1 + \theta)\log(1 + \theta)\big)$ for $\theta \in [0, 1]$, which can be simplified as:

$$R(f) - R^* \leq \frac{2}{\sqrt{2\pi_1 + \pi_2}}\sqrt{\overline{R}(f) - \overline{R}^*}.$$

## D. Calibration for unbounded below losses

In this section, we discuss the relationship between the unbounded below condition and calibration for surrogate loss functions. First, we present that the bounded below condition is indeed a necessary condition for classification-calibration.

**Corollary D.1.** *Suppose a convex loss function $\phi(\cdot)$ is unbounded below, then it is not classification-calibrated.*

*Proof.* Since $\phi(z)$ is convex and unbounded below, then $\phi(z) \to -\infty$ for $z \to \infty$. Let's consider the case where there exists a domain $\mathcal{X}_0$ such that $1 > \mathbb{P}(\mathbf{X} \in \mathcal{X}_0) > 0$ and $\eta(\mathbf{x}) = 1$ for $\mathbf{x} \in \mathcal{X}_0$. Then, define $f_l(\mathbf{x}) = l$ when $\mathbf{x} \in \mathcal{X}_0$, and $f_l(\mathbf{x}) = 0$ when $\mathbf{x} \notin \mathcal{X}_0$. In this case,

$$R_\phi(f_l) = \mathbb{E}\big(\mathbf{1}(\mathbf{X} \in \mathcal{X}_0)\phi(l)\big) + \mathbb{E}\big(\mathbf{1}(\mathbf{X} \notin \mathcal{X}_0)\phi(0)\big) \to -\infty = R_\phi^*,$$

however, it is clear that $R(f_l) \nrightarrow R^*$, since $f_l$ fails to match the Bayes rule when $\mathbf{x} \notin \mathcal{X}_0$. Thus, if $\phi$ is calibrated, it must be bounded below. $\qquad\square$

The concept of classification-calibration is considered "universal", as it requires the loss function to ensure consistency for all data distributions. We then investigate a weak version of calibration: can unbounded below loss functions guarantee calibration for some specific data distributions?

To address this question, we start by extending the if and only if condition for the equivalent definitions of classification-calibration in (Bartlett et al., 2006).

**Lemma D.2.** *For any loss function $\phi : \mathbb{R} \to (-\infty, \infty)$, $R_\phi^* > -\infty$ if and only if the statements (a) and (b) are equivalent, where statements (a) and (b) are defined as:*

   a. *For any $\eta \neq 1/2$, $H_\phi^-(\eta) > H_\phi(\eta)$.*

   b. *For every sequence of measurable function $f_l : \mathcal{X} \to \mathbb{R}$ and every probability distribution on $\mathcal{X} \times \{\pm 1\}$,*

$$R_\phi(f_l) \to R_\phi^* \quad \text{implies that} \quad R(f_l) \to R^*.$$

*Proof.* The necessity ($\Longrightarrow$). Suppose $R_\phi^* > -\infty$, then $(a) \implies (b)$ can be proved by following the proof of Theorem 1 (the last paragraph) in (Bartlett et al., 2006). Next, we tend to prove $(b) \implies (a)$ by contradiction. Suppose there exist a sequence of $\{f_l\}$ and some probability distribution, such that

$$R_\phi(f_l) \to R_\phi^*, \text{ but } R(f_l) \nrightarrow R^*. \tag{11}$$

Define

$$\Omega_\phi := \Big\{\mathbf{X} : \lim_{l\to\infty} C_\phi\big(\eta(\mathbf{X}), f_l(\mathbf{X})\big) = H_\phi\big(\eta(\mathbf{X})\big)\Big\},$$
$$\Omega := \Big\{\mathbf{X} : \lim_{l\to\infty} C_0\big(\eta(\mathbf{X}), f_l(\mathbf{X})\big) = H_0\big(\eta(\mathbf{X})\big)\Big\}.$$

Since $\phi(0) \geq R_\phi^* > -\infty$, and $1 \geq R^* \geq 0$, (11) implies that $\mathbb{P}(\Omega_\phi) = 1$ and $\mathbb{P}(\Omega) < 1$, and thus $\mathbb{P}(\Omega^c \setminus \Omega_\phi^c) \geq \mathbb{P}(\Omega^c) - \mathbb{P}(\Omega_\phi^c) > 0$. Therefore, there exists a subset $\mathcal{A} \subseteq \Omega^c \setminus \Omega_\phi^c$ with $\mathbb{P}(\mathcal{A}) > 0$, such that for any $\mathbf{x} \in \mathcal{A}$, we have

$$C_\phi(\eta, \alpha_l) \to H_\phi(\eta), \text{ but } C_0(\eta, \alpha_l) \nrightarrow H_0(\eta),$$

where $\eta := \eta(\mathbf{x})$ and $\alpha_l := f_l(\mathbf{x})$. This implies that $\exists \epsilon > 0, \forall L_0, \exists l > L_0$, such that, $C_0(\eta, \alpha_l) > H_0(\eta)$, and thus $\alpha_l(2\eta - 1) \leq 0$. Now, $H_\phi^-(\eta) = H_\phi(\eta)$ follows from by taking limits of $H_\phi(\eta) \leq H_\phi^-(\eta) \leq C_\phi(\eta, \alpha_l)$, and the fact that $C_\phi(\eta, \alpha_l) \to H_\phi(\eta)$, which contradicts to the calibration definition of $\phi$ in (a). This completes the proof of the necessity.

The sufficiency ($\Longleftarrow$). We construct the proof by contradiction. Suppose $R_\phi^* = -\infty$, we can find an example that (a) does not imply (b). Specifically, assume that $\phi$ is classification calibrated and $R_\phi(f_l) \to R_\phi^* = -\infty$, then there exists a sequence $\{f_l\}$ such that $\mathbb{P}(\Omega_\phi) = 1$, and a set $\mathcal{A}$ with $\mathbb{P}(\mathcal{A}) = c > 0$, such that $H_\phi(\eta(\mathbf{x})) = -\infty$, for $\mathbf{x} \in \mathcal{A}$. Now, we split $\mathcal{A}$ as two disjoint sets $\mathcal{A}_1$ and $\mathcal{A}_2$ each with probability measure $c/2$, and define a new sequence $\tilde{f}_l(\mathbf{x}) := f_l(\mathbf{x})$ for $\mathbf{x} \in \mathcal{A}_1^c$, and $\tilde{f}_l(\mathbf{x}) := 0$ for $\mathbf{x} \in \mathcal{A}_1$. In this case,

$$R_\phi(\tilde{f}_l) = \mathbb{P}(\mathbf{X} \in \mathcal{A}_1)\phi(0) + \mathbb{E}\Big(\big(\mathbf{1}(\mathbf{X} \in \mathcal{A}_2) + \mathbf{1}(\mathbf{X} \notin \mathcal{A})\big)C_\phi\big(\eta(\mathbf{X}), f_l(\mathbf{X})\big)\Big) \to -\infty = R_\phi^*.$$

However, $R(\tilde{f}_l) \nrightarrow R^*$ since $\mathbf{x} \in \mathcal{A}_1$, $C_0(\eta(\mathbf{x}), \tilde{f}_l(\mathbf{x})) = 1 > \min(\eta(\mathbf{x}), 1 - \eta(\mathbf{x})) = H_0(\eta(\mathbf{x}))$ and $\mathbb{P}(\mathcal{A}_1) = c/2 > 0$. This completes the proof. $\qquad \square$

Lemma D.2 relaxes the non-negativeness or bounded below condition for calibration in (Bartlett et al., 2006), which helps extend the if and only if conditions of classification-calibration to more general loss functions. For example, if we work with specific data distributions, many calibrated surrogate losses do not need to be unbounded below, as illustrated in the following examples.

**Hinge loss with varying right tails.** In this example, we examine the impact of various right tails on calibration of loss functions. Specifically, we adopt the shape of hinge loss function $\phi(z) = 1 - z$, when $z < 1$, and explore different right tails when $z \geq 1$, then discuss calibration.

- When $z \leq 1$, $\phi(z) = 1 - z$;

- When $z > 1$, (zero tail) $\phi(z) = 0$; (exponential tail) $\phi(z) = e^{-(z-1)} - 1$; (inverse tail) $\phi(z) = 1/z - 1$; (inv-log tail) $\phi(z) = e/\log(z + e - 1) - e$; (logarithm tail) $\phi(z) = -\log(z)$.

**Lemma D.3.** *Suppose $\eta(\mathbf{X}) \in [\epsilon, 1 - \epsilon]$ almost surely for any fixed $\epsilon > 0$, then the hinge loss, with the zero, exponential, inverse, inverse-logarithm, and logarithm tails, are all classification-calibrated.*

*Proof.* We can check that $R_\phi^* > -\infty$ for all pre-defined losses, and $\phi$ is convex with $\phi'(0) < 0$ then $\phi$ is classification-calibrated. $\qquad \square$

Therefore, some unbounded below loss functions can still provide calibration for particular data distributions. Unfortunately, unbounded below loss functions often exhibit instability during training with SGD, even if they are calibrated in simulated cases (see our GITHUB). One possible explanation is that the batch sampling may disrupt the distribution assumptions necessary for calibration of unbounded below losses. Conversely, bounded below calibrated loss functions do not rely on data distribution assumptions to maintain calibration.

This appendix delineates the relationship between the bounded below condition and calibration. First, the bounded below condition is a necessary condition for calibration. Next, by introducing additional distribution assumptions, it is demonstrated that certain unbounded below loss functions can still be calibrated. Lastly, through numerical experiments, it is suggested that unbounded below loss functions may exhibit potential instability during SGD training. In conclusion, the unbounded below condition (or superlinear raising tail) is identified as a critical yet often underestimated criterion for loss functions, both theoretically and in numerical implementations.

