# OpenReview forum: "EnsLoss: Stochastic Calibrated Loss Ensembles for Preventing Overfitting in Classification"
_ICML.cc/2025/Conference — ICML 2025 poster_

### Official Review · Reviewer_cfo1 · 2025-03-10

**Overall Recommendation:** 3

**Summary:**

This paper proposed a novel ensemble framework, EnsLoss, for mitigating overfitting issue for binary classification task. The EnsLoss is motivated by Calibration property studied in the literature and combined with the idea of ensemble, which makes the 'equivalent loss' also calibrated for ensemble training process. Excess risk bound and empirical experiments are provided to justify the proposed method.

## update after rebuttal
I keep the same score as weak acceptance as before. The proposed method looks novel to me. And there are some theoretical justification and empirical experiments for supporting the method. Given that there are still some space for further improving from i) multiclass setting; ii) more theoretical (or better) justifications; iii) more empirical baselines for comparison, I don't raise my score.

**Claims And Evidence:**

One claim that is not convincing to the reviewer is: the EnsLoss can mitigate overfitting issue.

Does EnsLoss enjoy better generalization error bound? Or is the excess risk bound better than fixed single loss?

Even for the empirical experiments, there is no demonstration that EnsLoss mitigates overfitting or simply fits the data better. Maybe empirically the training performance should also be presented for supporting the claim.

**Essential References Not Discussed:**

I didn't noticed any missing essential references.

**Experimental Designs Or Analyses:**

The experimental design and analyses generally make sense to me.

However, as mentioned previously, I don't see much connection and justification to 'mitigating overfitting issue'.

**Methods And Evaluation Criteria:**

Yes, the evaluation generally makes sense to me.

The only concern is that there is still a gap to the paper claim 'mitigate overfitting issue'.

**Other Comments Or Suggestions:**

Based on my previous comments:
1. It would be better for the paper to justify the claim 'mitigate overfitting issue' more clearly.
2. It would be more impactful to extend the method on multi-class classification setting.
3. It would be better to study/show the relations with existing ensemble methods, in order to fully demonstrate the impacts or limitations for the proposed method.

**Other Strengths And Weaknesses:**

Strengths:
1. The paper writing is well organized and the flow is clean.
2. The motivation is clear and the proposed ensemble method should be novel.
3. Some theoretical justification and empirical results are provided.

Weakness:
1. Neither the theoretical nor empirical results can clearly support that the proposed method can mitigate overfitting issue. There are still a gap to justify the claim (also see previous 'Claims And Evidence').
2. The binary classification application is restricted. Given that the multi-class classification also have been developed calibration theories in the literature, this work can enjoy higher impact if can be extended.
3. As a novel ensemble method, there is no comparison with other existing ensemble methods. We don't know whether the proposed ensemble method is better, nor if it is compatible with existing ensemble methods.
4. Minor concern: the fonts on Fig 1 and 5 are too small to be viewed clearly.

**Questions For Authors:**

Please see previous comments, thanks.

**Relation To Broader Scientific Literature:**

The paper contributes a novel ensemble method for binary classification motivated by Calibration property.

**Theoretical Claims:**

I didn't check the theoretical proof thoroughly in the appendix but the flow in the main contents looks correct.

---

> ### Author Rebuttal · Authors · 2025-03-29
>
> > (No theoretical evidence EnsLoss reduces overfitting - missing analysis of generalization error bounds or excess risk)
>
> **Reply.** Thank you for this insightful question. We do not explicitly provide the generalization bound, yet following your suggestion, we can show the advantages of *ensLoss* through the lens of Rademacher (Rad) complexity. The Rad complexity plays a crucial role in many concentration inequalities, so it will be helpful to infer the benefits in generalization bounds.
>
> The Rad complexity of $\phi$-classification is:
> $$
> \text{Rad}\_{\phi}(\mathcal{F}) := \sup\_{f \in \mathcal{F}}|\frac{1}{n} \sum\_{i=1}^n \tau_i \phi(Y_i f(\mathbf{X}_i))|,
> $$
> where $(\tau_i)$ are i.i.d. Rad random variables. The corresponding Rad complexity for ensLoss is:
> $$
> \overline{\text{Rad}}( \mathcal{F}) := \sup\_{f \in \mathcal{F}} |\frac{1}{n} \sum\_{i=1}^n \tau_i \mathbb{E}\_{\Phi} \Phi (Y_i f(\mathbf{X}_i) )| \leq \mathbb{E}\_{\Phi} (\text{Rad}\_{\Phi}(\mathcal{F})).
> $$
>
> Hence, Rad complexity of ensLoss is no worse than the average of a set of fixed losses, showing at least one theoretical benefit. Yet, it does not provide conclusive evidence of its superiority over fixed losses. We will honestly report the potential advantages and limitations in our revision.
>
> > (No numerical evidence EnsLoss reduces overfitting - training results should be shown alongside test results)
>
> **Reply.** We apologize for not including the training results. This was because almost all methods achieved a train accuracy of 1 (or close to 1), as demonstrated below for CIFAR2 (dog-cat) dataset.
>
> | Dataset  | Model    | Loss    | Train Acc | Test Acc |
> |----------|----------|---------|-----------|----------|
> | CIFAR2 | ResNet34 | ensLoss | 1.00 ± .00  | 0.72 ± .02 |
> |          |          | Hinge   | 1.00 ± .00  | 0.70 ± .01 |
> |          |          | BCE | 1.00 ± .00  | 0.69 ± .01 |
> |          |          | EXP     | 1.00 ± .00  | 0.60 ± .01 |
> |          | VGG16    | ensLoss | 1.00 ± .00  | 0.81 ± .01 |
> |          |          | Hinge   | 1.00 ± .00  | 0.78 ± .01 |
> |          |          | BCE | 1.00 ± .00  | 0.77 ± .01 |
> |          |          | EXP     | 1.00 ± .00  | 0.69 ± .00 |
>
> Moreover, in Table 6, the accuracy appears to decrease when the model becomes more complex (e.g., from VGG16 to VGG19). This is also a typical phenomenon associated with overfitting.
>
> Therefore, we reasonably conclude the improvement by *ensLoss* is mitigates overfitting not simply fits the data better.
>
> > (Limited to binary classification - extend to multiclass where calibration theories exist)
>
> **Reply.** Thank you for this valuable comments. We agree that extending our work would be beneficial. Our current focus on binary classification is deliberate for several reasons:
>
> First, the if-and-only-if condition for calibration in binary case is well-studied and straightforward to implement, allowing us to directly validate our ensemble idea without additional complications.
>
> Second, as indicated in Theorem 7 of [1], the iff calibration conditions for multiclass case are substantially more complex. Implementation would require additional specifications in the formulations and sufficient conditions for calibration, as shown in Section 4 of [2] and Section 5 of [1]. While extension is possible using simplified conditions and specific formulations (Prop 15, [1]), yet there remains ongoing debate regarding optimal formulations. Binary case allows us to avoid these debates and directly compare the core ensemble idea.
>
> Finally, binary classification itself is a fundamentally important problem with many applications. We believe that improvements are also highly significant.
>
> **References:**
> 1. Tewari & Bartlett (2007). On the Consistency of Multiclass Classification Methods. JMLR.
> 2. Zhang (2004). Statistical analysis of some multi-category large margin classification methods. JMLR.
>
> > (No comparison with existing ensemble methods)
>
> **Reply.** Agree. To illustrate, we report test performance for model averaging (m-ave) and majority voting (m-vote) on CIFAR2 (dog-cat) using ResNet101 over three fixed losses:
>
> ||Acc|AUC|
> |-|-|-|
> |BCELoss|67.33(.23)|73.35(.24)|
> |EXP|53.62(.48)|54.86(.35)|
> |Hinge|67.12(.32)|68.60(.66)|
> |**m-ave**|68.14(.25)|74.35(.26)|
> |**m-vote**|67.93(.26)|---|
> |ensLoss|**70.07(.90)**|**76.60(.87)**|
>
> As shown, ensembling indeed improves single fixed loss methods, but enLoss still performs better. We observed these conclusions across datasets and will add details in the revision.
>
> The challenge with ensembling methods is that selecting too many fixed losses requires substantial computational costs of training, while combining too few losses produces less effects. A key advantage of ensLoss is that it requires only a single model training.
>
> > (the fonts on Fig 1 and 5 are too small)
>
> **Reply.** Thank you for comments. We will revise them in the revision.

---

### Official Review · Reviewer_ecqi · 2025-03-11

**Overall Recommendation:** 3

**Summary:**

This paper considers the problem of binary classification, where it proposes a method called EnsLoss that combines with stochastic gradient descent (SGD) to obtain the optimization objective to train a classifier. The key idea of EnsLoss is based on the convenient classification-calibrated condition (Bartlett+, 2006) that the derivative at 0 is negative for a bounded-from-below convex loss is sufficient to state that a loss is calibrated. Based on this information, Ensloss tries to obtain a reasonable derivative that gives rise to a classification-calibrated loss. Theoretical analysis of classification and excess risk bound confirms the validity of the proposed method. Experiments show that with SGD, the proposed Ensloss outperformed using a fixed loss function (e.g., Logistic, Hinge, EXP) significantly, while it can also combine with other regularization schemes (e.g., dropout).

## update after rebuttal
I have read the rebuttal and appreciate the authors' response to my concerns. I think this paper has merits that outweigh its flaws, and I want to keep my score on the acceptance side. The contribution is limited to binary classification is acceptable to me as it is an important problem setting, and I feel the extension to the multi-class might require a further significant amount of work as the classification calibration condition is not as simple as binary classification.

**Claims And Evidence:**

1. Methodologically, the proposed method is clear and convincing.
2. The experimental results showed superiority of the proposed method. But I still feel there is no clear explanation why this method is better than a fixed loss. Since a fixed loss can also be classification-calibrated and has a reasonable excess risk bound.

**Essential References Not Discussed:**

I do not have any additional suggestions.

**Experimental Designs Or Analyses:**

1. Experimental results are reasonable.
2. Unfortunately, it would be nice to have more ablation studies to highlight the importance of each condition, e.g., whether Line 13 is important (bounded below). In the empirical sense, I am not sure if this is needed, as function class is already somewhat bounded in practice.

**Methods And Evaluation Criteria:**

1. Many datasets were used in the experimental result section. The evaluation criteria: accuracy and AUC (in the appendix) are reasonable.
2. This paper kindly provides the section of compatibility of the proposed method with other regularization methods, which nicely shows the usefulness of Ensloss.

**Other Comments Or Suggestions:**

Overall, I think the paper is easy to follow.

Suggestions on writing:
1. Please include the discussion of Algorithm 2 if possible as it is important for understanding Algorithm 1. Or try to rewrite Algorithm 1 in a way that doesn't rely too much on Algorithm 2.
2. I know dropout is very popular and common, but I think in the discussion of related work, there is no description at all how it works, i.e., randomly deactivates a subset of neurons during training.
3. What is Assumption 1 in Theorem 3.6? Do you mean Assumption 3.4? Assumption 1 is also referred to in the appendix section.

Current score before the rebuttal:
Overall, I think the idea to make classification-calibrated loss without directly defining a loss for binary classification is interesting. The proposed method is reasonable and achieves good experimental results. I listed some weaknesses in the weaknesses section above.

**Other Strengths And Weaknesses:**

Strengths
1. Theoretically justified.
2. Easy to implement under SGD framework.
3. Paper is easy to follow.
4. Many datasets were used in the experimental results, which I found impressive.

Weaknesses
1. Limited to binary classification
2. Limited to SGD optimization algorithm
3. The algorithm seems to require more training time to stabilize the training accuracy as noted in the conclusion section.
4. I still feel it is unclear why Ensloss can significantly outperform logistic loss and hinge loss.

**Questions For Authors:**

1. I still feel it is unclear why Ensloss can significantly outperform logistic loss and hinge loss. Could you explain intuitively why this is the case?
2. Do exp, hinge, logistic, squared losses satisfy superlinear raising tail condition? If not, then is there any well-known fixed loss that satisfies this condition?
3, What is z (not bold) in line 15, I failed to follow this part. Is it a constant for all data points in the mini-batch?
4. Regarding bounded below condition, I believe it might not be a big concern if we assume f is bounded. For example, there is an unhinged loss which is simply a linear loss $\phi(z)=1-z$ that has robustness and is classification-calibrated if function is bounded (proposition 5: Learning with Symmetric Label Noise: The Importance of Being Unhinged: https://arxiv.org/abs/1505.07634). Do you find this condition highly important in the experiment?
5. Can we know the form of the risk minimizer of the Ensloss? For example, hinge loss risk minimizer will be sign(p(y=1|x)-0.5), 2p(y=1|x)-1 for squared loss.

**Relation To Broader Scientific Literature:**

It is called Ensloss, but I also feel it is more like a design of objective that bypasses the loss design but goes straight to the derivative design for using stochastic gradient descent. Perhaps additional discussion on derivative-design for SGD for several problems might be nice.

**Theoretical Claims:**

1. I briefly checked the proof and think the result is reasonable.

---

> ### Author Rebuttal · Authors · 2025-03-30
>
> > (No clear explanation of why EnsLoss outperforms fixed losses)
> >
> > (Intuitive explanation)
>
> **Reply.** We appreciate the opportunity to clarify why ensLoss outperforms fixed losses.
> 1. An intuitive explanation:
> - ensLoss is a method conceptually similar to Dropout, offering benefits through ensemble effects.
> - More importantly, for fixed losses, the only truly "necessary" condition is calibration, with potential overfitting arising from extra structures (e.g., hinge loss pushes classifiers to exactly +1/-1). ensLoss *not* require to satisfy a specific risk minimizer, instead seeking a good classifier across various random losses while maintaining calibration. This can be considered a form of *regularization*—we avoid pushing the classifier to excessively satisfy any particular loss, thereby reducing overfitting.
>
> 2. While calibration and excess risk bounds are crucial to verify consistency, they are not necessarily informative about comparative performance on specific datasets. Distribution-free upper bounds are often not tight under particular datasets, making it difficult to determine which loss will perform better on a given dataset.
>
> 3. We add theoretical support for ensLoss through Rademacher (Rad) complexity. The Rad for ensLoss is:
> $$
> \overline{\text{Rad}}(\mathcal{F}):=\sup\_{f \in \mathcal{F}}|\frac{1}{n}\sum\_{i=1}^n \tau_i \mathbb{E}_{\Phi} \Phi(Y_i f(\mathbf{X}_i) )|\leq \mathbb{E}\_{\Phi}(\text{Rad}\_{\Phi}(\mathcal{F})),
> $$
> where $\tau_i$ are i.i.d. Rad random variables, and $\text{Rad}\_{\Phi}(\mathcal{F})$ is the Rad of a fixed $\phi$-classification. Hence, Rad complexity of ensLoss is no worse than the average of a set of fixed losses, showing at least one theoretical benefit.
>
> > (Ablation studies on "bounded below")
>
> **Reply.** We greatly appreciate this comment. This is precisely an important step when developing ensLoss: we also initially overlooked this condition, but discovered that is indeed critical.
>
> To demo its effect, we reported typical train/test Acc for CIFAR2 (dog-cat) using VGG16 w/wo the bounded below (BB) condition (showing select epochs).
> |Epoch|w/o BB||w/ BB||
> |-|-|-|-|-|
> ||Train Acc|Test Acc|Train Acc|Test Acc|
> |20|0.50|0.49|0.96|0.74|
> |60|0.50|0.51|1.00|0.83|
> |120|0.50|0.49|1.00|0.83|
> |160|0.51|0.49|1.00|0.83|
> |200|0.50|0.49|1.00|0.83|
>
> *Finding*
> - w/o: model fails to learn (random-level train accuracy)
> - w/: effective learning and prediction
>
> *Explanation*
>
> From a loss-derivative perspective, Line 13 discounts gradients for correctly classified samples ($z=yf(x)>1$), focusing optimization efforts on misclassified/boundary points—a standard loss design principle.
>
> > (Limited to binary classification)
>
> **Reply.** Agree, we will discuss the limitation and potential extensions, ensLoss can be extended to more general ML tasks with calibration conditions. Also, we focus on BC because:
> - The iff condition for calibration in BC is well-studied and straightforward to implement, allowing us to directly validate our ensemble idea without additional complications.
> - BC itself is a fundamentally important problem with tremendous applications. We believe that improvements are also highly significant.
>
> > (ensLoss requires longer time for stabilization)
>
> **Reply.** We agree this limitation and provide precise time comparisons. We report the *min epoch* for stabilization (after which train acc remains within an error margin of 0.005) on CIFAR2 (dog-cat).
> ||BCE|Hinge|ensLoss|
> |-|-|-|-|
> |ResNet101|80|60|160|
> |ResNet34|70|25|145|
> |VGG16|15|25|55|
> |VGG19|45|35|150|
>
> As indicated, ensLoss typically requires 2-3x more epochs to stabilize, which remains acceptable.
>
> > (Discussion on derivative-design for SGD)
> >
> > (Limited to SGD)
> >
> > (Suggestions on writing)
> >
> > (typos of $z$ in Algo 1)
>
> **Reply**. Agree, we will: (1) move Algo 2 into the main text; (2) add a description of dropout; (3) discuss derivative-design methods; (4) discuss the limitations of relying on SGD; and (5) correct typos: such as replacing $z$ with $\gamma$.
>
> > (Superlinear for existing losses)
>
> **Reply.** All the mentioned losses (exp, hinge, logistic, and square) satisfy the superlinear tail condition (proofs omitted due to space constraints).
>
> > (The "bounded below" may be unnecessary if f is bounded)
>
> **Reply.** Yes, if $f$ is bounded, the condition can be relaxed.
>
> In fact, for a convex loss, boundedness of $f$ is a sufficient condition of boundedness of $\phi$. Assume $|f(x)| \leq B$, $|z| = |y f(x)| \leq B$. Since $\phi$ is convex, it is continuous, thus $\phi$ is bounded for $|z| \leq B$.
>
> > (risk minimizer of the Ensloss)
>
> **Reply.** Thank you for this insightful question. Unlike fixed losses, ensLoss has no exact form of risk minimizer, as the minimizer would depend on the distribution of the underlying losses. Yet, as indicated in Theorem 3.6, the method is calibrated, ensuring that the minimizer of ensLoss shares same sign as $P(Y=1|x) - 0.5$, although we cannot determine its exact form.

---

> > ### Comment · Reviewer_ecqi · 2025-04-03
> >
> > I would like to thank the authors for providing detailed responses to my concerns.
> >
> > 1. Intuitive explanation on Ensloss.
> > I appreciate the intuitive explanation of Ensloss: The flexibility of the optimal solution function form of Ensloss minimizer might make it easier to optimize accuracy **than** a fixed loss function, where a risk minimizer form is usually fixed.
> >
> >    Regarding Rademacher complexity of using Ensloss is smaller or at least as large as using a fixed loss. I found it strange to say something like the lower Rademacher complexity indicates a better result. If we only compare Rademacher complexity, then what we can say is that the function class f learned with Ensloss is less complex than using a fixed loss, hence less flexible. Perhaps we need to directly compare the estimation error bound or something like that to say the advantage.
> >
> > 2. Bounded below condition:
> > I was surprised that this condition is so critical. Thanks for the result!
> >
> > 3. Slower than fixed loss:
> > I think it is a good idea to clearly write (which the submission already did) in the paper that the disadvantage is that Ensloss training is slower to finish. I think this is not a big concern, and one may try to optimize it in future work.
> >
> > I am still leaning toward the accept side I think I did not have a misunderstanding in this paper. I will incorporate this information in the discussion phase with other reviewers.

---

> > > ### Author Response · Authors · 2025-04-03
> > >
> > > Thank you for your insightful and careful review as well as your prompt reply!
> > >
> > >
> > > We apologize that due to the word limit in the rebuttal, we could not clearly express and explain the results of Rademacher complexity. We would like to take this opportunity to provide more details and clarify further.
> > >
> > >
> > > When comparing the Rademacher complexities for *ensLoss* and fixed losses, the functional class $\mathcal{F}$ used in both cases is the same; the differences in Rademacher complexities arise solely from the *different loss functions*. With a fixed loss $\phi$, the Rademacher complexity is defined as
> > > \begin{equation*}
> > >   \text{Rad}\_{\phi}( \mathcal{F} ) := \sup\_{f \in \mathcal{F}} \Big|  \frac{1}{n} \sum\_{i=1}^n \tau\_i \phi\big(Y\_i f(\mathbf{X}\_i) \big) \Big|,
> > > \end{equation*}
> > > where $(\tau_i)_{i=1}^n$ are i.i.d. Rademacher random variables independent of $(\mathbf{X}\_i, Y\_i)\_{i=1}^n$.
> > >
> > > In contrast, the corresponding Rademacher complexity for the proposed ensemble loss method is:
> > > \begin{equation}
> > >   \overline{\text{Rad}}\_{\text{ens}}( \mathcal{F} ) := \sup\_{f \in \mathcal{F}} \Big| \frac{1}{n} \sum\_{i=1}^n \tau\_i \mathbb{E}\_{\Phi} \Phi\big(Y\_i f(\mathbf{X}\_i) \big) \Big| \leq \mathbb{E}\_{\Phi} \big( \text{Rad}\_{\Phi}(\mathcal{F}) \big).
> > > \end{equation}
> > > This inequality suggests that the complexity of the ensemble method is not worse than the average of the complexities corresponding to the fixed losses in the family.
> > > According to Theorem 6.4 in [1], a lower Rademacher complexity improve the upper bounds of the estimation error of the empirical risk minimization method.
> > >
> > > We absolutely agree with your observation that comparing complete estimation error bounds would yield a more convincing results. In our revised manuscript, we will include estimation error bounds to further discuss the potential benefits of *ensLoss* while also offering a transparent discussion of the limitations of our theory.
> > >
> > > We hope our reply and rebuttal will enhance our paper and provide necessary explanation and clarification. We respectfully hope you might consider these points in your final assessment. Thank you again for your thoughtful review!
> > >
> > > **Reference**
> > >
> > > [1] Zhang, T. (2023). Mathematical Analysis of Machine Learning Algorithms. Cambridge University Press.

---

### Official Review · Reviewer_UHGu · 2025-03-13

**Overall Recommendation:** 2

**Summary:**

The paper introduces EnsLoss, a novel ensemble learning method that applied the idea of ensemble to loss functions during model training within the empirical risk minimization (ERM) framework. Specifically, instead of explicitly using one loss functions, the author proposes to randomly sample loss functions on the fly during ERM. In practice, the authors propose transforming convexity and calibration conditions into loss-derivatives, allowing for direct generation of calibrated loss-derivatives. The authors verifies the validity of the proposed method through theoretical analysis. They also conduct experiments across multiple datasets and deep learning architectures to demonstrate its superiority over fixed loss methods.

**Claims And Evidence:**

I think the authors of the paper provides some good theoretical justification to the proposed method. However, I feel that empirical verification is weak. Specifically, according to Table 4, the authors of the paper only conduct experiments with CIFAR2, which is puzzling to me. Why don't we use CIFAR10? Moreover, the accuracies of the CIFAR2 results seems rather low. For instance, according to Table 6, the accuracy of CIFAR2 with ResNet34 is approximately 76%. Prior literature has shown that similar architecture achieves 90%+ accuracy on the CIFAR10 dataset. What is causing the huge gap? As such, I am not confused that the proposed method is preferred over BCE loss.

**Essential References Not Discussed:**

I think the authors needs additional discussion about different loss functions proposed to train neural networks. While the idea of dynamically sample loss functions is new ,a lot of prior works have been proposed that changes the gradients of the loss functions. Some of the examples include:

1. Lin, Tsung-Yi, et al. "Focal loss for dense object detection." Proceedings of the IEEE international conference on computer vision. 2017.
2. Leng, Zhaoqi, et al. "Polyloss: A polynomial expansion perspective of classification loss functions." arXiv preprint arXiv:2204.12511 (2022).
3. Zhang, Zhilu, and Mert Sabuncu. "Generalized cross entropy loss for training deep neural networks with noisy labels." Advances in neural information processing systems 31 (2018).
4. Ma, Xingjun, et al. "Normalized loss functions for deep learning with noisy labels." International conference on machine learning. PMLR, 2020.

**Experimental Designs Or Analyses:**

Yes. As argued above, I am not confident about the experiments conducted by the authors.

**Methods And Evaluation Criteria:**

As argued above, I don't think the evaluation criteria makes sense for the problem.

**Other Comments Or Suggestions:**

N.A.

**Other Strengths And Weaknesses:**

I think the idea of randomly sampling loss functions, while incremental arguably, is interesting. However, I am not fully convinced why, both theoretically and empirically, that the proposed loss function is better.

**Questions For Authors:**

Why do we want a loss ensemble instead of a fixed loss function from a theoretical standpoint? Any benefits that we can show?

**Relation To Broader Scientific Literature:**

The proposed method is a general loss function that can be applied to training of all sort of machine learning models.

**Theoretical Claims:**

I briefly went over the claims, but did not have time to go through all the proofs provided in the appendix. In general, the claims makes sense theoretically, and seems technically sound.

---

> ### Author Rebuttal · Authors · 2025-03-29
>
> > (Empirical verification weak - only tested on CIFAR2, not CIFAR10.)
>
> **Reply.** To clarify, our current work focuses specifically on *binary classification*. Since CIFAR10 is a multi-class dataset with 10 categories, we derived binary classification problems from it by creating all possible pairwise combinations of the original classes, which resulted in 45 distinct CIFAR2 datasets (calculated as ${10}\choose{2}$ = 45 pairs).
>
> > (The accuracies of the CIFAR2 results seems rather low compared to CIFAR10 in literature)
>
> **Reply**. The performance gap you've noted stems from several differences between CIFAR10 and CIFAR2:
>
> 1. **Dataset size**: Each of our CIFAR2 datasets contains only 10,000 training images (5,000 from each of the two selected classes), whereas the full CIFAR10 dataset contains 50,000 training images. This significant reduction in training data naturally impacts model performance.
>
> 2. **Classification difficulty variation**: As shown in Figure 4 of our manuscript, the accuracy of CIFAR2 varies substantially depending on which pair of classes is selected, ranging from 68% to 96%. Some binary classification tasks (e.g., distinguishing between cats and dogs) are inherently more challenging than others (e.g., distinguishing between automobile and horse).
>
> Therefore, the 90%+ accuracy for CIFAR10 is a 10-class classification task with full training data. Our experimental setup is fundamentally different, making direct accuracy comparisons is not meaningful.
>
> In addition, our experimental design aims to provide a fair comparison across different loss functions and the proposed ensLoss. The implementation settings for each method are identical, with the only difference being the loss functions used. Additionally, we have provided anonymous GitHub open source code to enable replication of our results.
>
> > (Algorithm 2 from Appendix A should be in main text for better understanding)
>
> **Reply**. Thank you for the suggestion. In our revision, we will move Algorithm 2 and the relevant explanatory content from Appendix A into the main text, specifically after Section 3.2 before Section 3.3 where we introduce our methodology.
>
> > (Need discussion of prior works on modified loss functions/gradients for neural networks)
>
> **Reply.** Thank you for highlighting these important references on loss function modifications. We agree that our work could benefit from being positioned within the broader context of gradient-based loss function innovations.
>
> The papers you've mentioned represent significant contributions to loss function design. Focal Loss addresses class imbalance by down-weighting well-classified examples. PolyLoss provides a unified framework through polynomial expansions. Generalized Cross Entropy and Normalized Loss Functions tackle noisy label scenarios through robust formulations. As you mentioned, our approach differs in that we dynamically sample from a collection of loss functions rather than proposing a fixed modified loss.
>
> In our revision, we will expand Section 2 to include these references and others, providing a more comprehensive discussion of loss function design from a gradient perspective.
>
> > (Unconvinced about theoretical and empirical advantages of randomly sampling loss functions)
>
> **Reply.** For empirical evidence demonstrating superior of *ensLoss*, please refer to our reply to the first two points, which clarifies why CIFAR10 (and its performance) is not appropriate to be used or referenced in binary classification.
>
> For theoretical justification, we supplement the theoretical advantages of *ensLoss* through the lens of Rademacher complexity. The Rademacher complexity plays a crucial role in determining the convergence rate of the excess risk (with smaller values yielding a faster rate).
>
> Specifically, given a classification function space $\mathcal{F}$, the Rademacher complexity of $\phi$-classification are defined as follows:
> $$
> \text{Rad}_{\phi}( \mathcal{F} ) := \sup\_{f \in \mathcal{F}} | \frac{1}{n} \sum\_{i=1}^n \tau_i \phi (Y_i f(\mathbf{X}_i) ) |,
> $$
>
> where $\tau_i$ are i.i.d. Rademacher random variables. On this ground, the corresponding Rademacher complexity for the proposed ensemble loss method can be formulated as:
> $$
>   \overline{\text{Rad}}( \mathcal{F} ) := \sup\_{f \in \mathcal{F}} | \frac{1}{n} \sum\_{i=1}^n \tau_i \mathbb{E}_{\Phi} \Phi (Y_i f(\mathbf{X}_i) ) | \leq \mathbb{E}\_{\Phi} ( \text{Rad}\_{\Phi}(\mathcal{F}) ),
> $$
>
> where the inequality follows from the Jensen's inequality.
>
> This simple deduction yields a positive result, that is, the Rademacher complexity of the ensemble loss is no worse than the average based on the set of fixed losses. Notably, identifying an effective fixed loss is often challenging due to the varying data distributions across different datasets, which highlights the potential of the ensemble loss as a promising solution.

---

> > ### Comment · Reviewer_UHGu · 2025-04-04
> >
> > I would like to thank the authors for providing me with feedback. I have read through the rebuttal and along with other reviewer's comments. While my concerns regarding empirical results were partially resolved, I am still not fully convinced how broadly applicable the proposed loss is, since it only works for binary classification. I have updated my score accordingly.

---

> > > ### Author Response · Authors · 2025-04-05
> > >
> > > Thank you for your feedback and for improving the rating of our paper.
> > >
> > > Regarding the limitations of our work in binary classification and its potential extension to more general machine learning problems, we would like to provide additional information to explain these points.
> > >
> > > **Potential extension**
> > >
> > > The primary motivation behind our paper consists of two components: "*ensemble*" and the "*calibration*" of the loss functions. Therefore, this concept can be applied to various ML problems, by identify the appropriate consistency/calibration conditions for different tasks. Fortunately, some consistency conditions have been extensively studied in the literature, including Section 4 in [1] and Theorem 1 in [2] for multi-class classification, Theorem 2 in [3] for bipartite ranking or AUC optimization, and Theorem 3.4 in [4] for asymmetric classification.
> > >
> > > Having demonstrated the feasibility of our approach in binary classification, these established consistency conditions can be incorporated into our ensemble loss framework, presenting promising directions for future research and empirical validation. We will elaborate on these extensions in our revised manuscript.
> > >
> > > **Why binary classification**
> > >
> > > We would also like to explain why we currently focus on binary classification:
> > >
> > > First, the if-and-only-if condition for calibration in binary classification is well-studied and straightforward to implement, allowing us to directly validate our ensemble approach without additional complications. We believe that is a natural proof of concept before extending to more complex ML tasks.
> > >
> > > Second, despite its apparent simplicity, binary classification remains a fundamental problem with significant methodological importance and practical relevance. It serves as a building block for many modern machine learning applications. We believe that methodological advancements in binary classification have substantial value to the field and broad implications for related problems.
> > >
> > > We respectfully hope you might consider these extension possibilities in your final assessment. While we acknowledge the current binary classification focus, and we hope our contributions in this fundamental setting can be valued. Thank you again for your feedback.
> > >
> > >
> > > **References**
> > >
> > > [1] Zhang, T. (2004). Statistical behavior and consistency of classification methods based on convex risk minimization. The Annals of Statistics, 32(1), 56-85.
> > >
> > > [2] Zou, H., Zhu, J., & Hastie, T. (2008). New multicategory boosting algorithms based on multicategory fisher-consistent losses. The Annals of Applied Statistics, 2(4), 1290.
> > >
> > > [3] Gao, W., & Zhou, Z. H. (2012). On the consistency of AUC pairwise optimization. IJCAI (pp. 939-945), 2015.
> > >
> > > [4] Scott, C. (2012). Calibrated asymmetric surrogate losses. Electronic Journal of Statistics, 6:985-992, 2012.

---

### Official Review · Reviewer_zDea · 2025-03-18

**Overall Recommendation:** 3

**Summary:**

This paper introduces EnsLoss,  a stochastic ensemble learning method specifically designed to mitigate overfitting in classification tasks. The main idea is is to ensemble different surrogate loss functions during SGD. The authors provide theoretical analysis and empirical results across various datasets. The results indicate that EnsLoss consistently outperforms fixed-loss methods, particularly for overparameterized models like ResNet.

**Claims And Evidence:**

Strenghts:
1. The authors provide empirical results that show consistent improvements over fixed losses.
2. EnsLoss is compatible with prevent-overfitting methods (section 4.4).

Weaknesses:
1. The alrogithm/practial experiments and theoretical framework (some conditions and proof) mismatch, see detailed questions in Theoretical Claims.
2. Limited scope of experiments (only binary classification, and small-scale image and tabular datasets) and also need to provide more ablation studies and baselines, see detailed questions in Methods And Evaluation Criteria and Experimental Designs Or Analyses.

**Essential References Not Discussed:**

N/A

**Experimental Designs Or Analyses:**

Weaknesses:
1. The authors choose $p=1$ for simplicity in the actual implementation in line 263, but in Lemma 3.2 the authors propose taking $g(z)=1/z^p$ with $p>1$. It may lead the loss unbounded below.
2. The authors mentioned "Another benefit of ENSLOSS is its relative insensitivity from time-consuming hyperparameter tuning" in Lines 405-406, but the $\lambda$ is still a hyperparameter affecting the results (Table 5).
3. Since authors are inspired by ensemble ideas, there should be more baselines to be compared, such as ensembling fixed loss
4. Longer training with traditional loss functions might mitigate overfitting similarly, so it is better to provide more detailed analysis on whether EnsLoss truly provides advantages beyond simply running more epochs.

**Methods And Evaluation Criteria:**

Strenghts:
1. The conditions for loss derivatives are derived from established theory
2. The ensemble risk framework (Theorem 3.6) connects the proposed method to statistical consistency.
3. It is pratical that the EnsLoss can be compatible with exisinting SGD based pipelines.

Weaknesses:
1. No ablation studies on Inverse Box-Cox transformation (9 in Algorithm 1)
2. All experiments only focus on binary classification, which limits the scope of the proposed methods.
3. The authors mentioned the limition of EnsLoss requiring more epochs in general, but there is no evaluations on computational costs.

**Other Comments Or Suggestions:**

If retaining $p=1$, it is better to provide additional explicit conditions or constrants on the random loss derivatives, or clearly state the theoretical limitation of Lemma 3.2

**Other Strengths And Weaknesses:**

N/A

**Questions For Authors:**

N/A

**Relation To Broader Scientific Literature:**

EnsLoss extends the idea of dropout to the loss space while grounding it in calibration theory. However, the authors only conduct experiments on binary classficiation, which leave open questions about its broader applicability.

**Theoretical Claims:**

Weaknesses:
1. The proof of Theorem 3.6 assumes a finite loss space via Lemma 3.5, but the algorithm generates loss derivatives stochastically which implys an infinite space. The authors argue that fixing the number of epochs makes finite in practice, but this is not formally proven.
2. The inverse Box-Cox transformation in Algorithm 2 can generate gradients from exponential distributions, so produces unbounded gradients, which violates the condition 3 in Assumption 3.4.
3. The theoretical framework assumes pre-defined losses (possibly random but static) as Eq 4, while the algorithm dynamically generates losses via SGD. The stochasticity of loss sampling is not addressed in the proofs.

---

> ### Author Rebuttal · Authors · 2025-03-30
>
> > (No ablation studies on invBC transformation)
>
> **Reply.** Thank you for the comment. Table 5 partially shows our invBC ablation studies: (1) Fixed case ($\lambda=0$) used in main experiments, where invBC is somewhat "deactivated" with gradients sampling from a fixed log-normal distribution; (2) Dynamic case with varying $\lambda$ sampled at different periods, where invBC is activated. This comparison reveals invBC's impact.
>
> We agree these results are insufficient and will include expanded analysis in our revision.
>
> > (Limited to binary classification)
>
> **Reply.** We agree and will discuss that ensLoss can be extended to more ML tasks with calibration conditions.  We focused on binary classification (BC) because:
> 1. The if-and-only-if condition for calibration in BC is well-studied and straightforward to implement, allowing us to directly validate our ensemble idea without additional complications.
> 2. BC itself is a fundamentally important problem with many applications. We believe that improvements are also highly significant.
>
> > (No evaluation of computational costs of ensLoss with additional epochs)
>
> **Reply.** Agree. We now provided time comparisons: report the *min epoch* required for training to stabilize (after which train accuracy remains within an error margin of 0.005) on the CIFAR2 (cat-dog) as a demo.
> ||BCE|Hinge|ensLoss|
> |-|-|-|-|
> | MobileNet|90|40|110|
> | MobileNetV2|80|35|90|
> | ResNet101|80|60|160|
> | ResNet50|45|35|150|
> | VGG16|15| 25|55|
> | VGG19|45|35|150|
>
> As indicated, *ensLoss* training typically requires 2-3x more epochs to stabilize, which remains acceptable.
>
> > (Theorem 3.6 assumes finite loss space via Lemma 3.5)
>
> **Reply.** To clarify, Theorem 3.6 relies solely on Assumption 3.4 (A3.4) and NOT on finite loss space. Lemma 3.5 provides only a sufficient condition for A3.4.
>
> E.g., a loss space $\phi(z)=(1-\beta z)\_+$ with $\beta \sim$ U(1,2), contains infinitely many losses, yet satisfies A3.4. We will clarify it in our revision to prevent confusion.
>
> > (invBC can produce unbounded gradients)
>
> **Reply.** Thanks for this key insight. In our implementation, we applied clipping at -1 for the loss-derivatives (Line 46 in `loss.py`). This was done initially because BCE and hinge gradients are both bounded by 1, ensuring fair comparison by eliminating any potential effect from differences in loss-derivative magnitudes. We will add the clipping step between Lines 9-10 in Algo 1.
>
> > (Theory assumes static losses while algo uses dynamic losses)
> >
> > ($p=1$ is used but theory requires $p>1$)
>
> **Reply.** This concern may be related to the finite loss space assumption mentioned earlier. Our framework (4) requires only that losses satisfies Assumption 3.4, not enumeration of all possible losses.
>
> For our implementation, Algo 1's generated loss-derivatives naturally satisfy conditions 1 and 3.
>
> We agree a small gap in condition 2 between theory $(p > 1)$ and algo $(p=1)$, but it really has a negligible effect on numerical results. E.g., using $p=1+10^{-10}$ would satisfy theoretical conditions with negligible numerical difference, as $z^{10^-10}$ falls beyond `float32` precision even for huge $z$ values.
>
> > (About insensitivity of tuning)
>
> **Reply.** We appreciate the careful reading. To clarify,
> - We used fixed $\lambda=0$ in all main experiments, achieving strong performance without tuning.
> - Table 5 merely explores invBC effects, not suggesting necessary tuning. Unlike methods requiring careful adjustment of parameters, *ensLoss* needs minimal tuning effort.
>
> We will clarify that Table 5 is for ablation study of invBC not suggesting tuning necessity.
>
> > (Comparison with ensembling fixed losses)
>
> **Reply.** Agree. To illustrate, we report test performance for model averaging (m-ave) and majority voting (m-vote) on CIFAR2 (dog-cat) using ResNet101 over three fixed losses:
>
> ||Acc|AUC|
> |-|-|-|
> |BCELoss|67.33(.23)|73.35(.24)|
> |EXP|53.62(.48)|54.86(.35)|
> |Hinge|67.12(.32)|68.60(.66)|
> |**m-ave**|68.14(.25)|74.35(.26)|
> |**m-vote**|67.93(.26)|---|
> |ensLoss|**70.07(.90)**|**76.60(.87)**|
>
> As shown, ensembling indeed improves single fixed loss methods, but enLoss still performs better. We observed these conclusions across datasets and will add details in the revision.
>
> The challenge with ensembling methods is that selecting too many fixed losses requires substantial computational costs of training, while combining too few losses produces less effects. A key advantage of ensLoss is that it requires only a single model training.
>
> > (Need evidence that EnsLoss offers advantages beyond just running more epochs)
>
> **Reply.** We clarify that all methods used the same number of epochs. We also provided epoch-level comparisons, as shown in Fig 1, where performance stabilizes after 100-150 epochs, with ensLoss maintaining its advantage. This pattern appeared across all datasets. We'll include more results in our revision. The persistent gap confirms ensLoss benefits aren't simply from long-epoch training.

---

### Decision · Program_Chairs · 2025-05-01

**Decision:**

Accept (poster)

**Comment:**

This paper extends classification-calibrated losses by allowing the "stochasticity" in the choice of loss functions. In other words, a binary classification can be a random variable so that we generate its gradient (based on the Box-Cox transform) in backward. This can be thought as the "ensemble" of loss functions. The authors provided conditions to ensure this loss is classification-calibrated, and extensive experiments to demonstrate superior performances over conventional "non-ensemble" loss functions.

During the author-reviewer discussions, there were several important discussions sparked. Specifically, some reviewers argued that it is restrictive if we can only apply to binary classification; others argued that it is theoretically unclear why the ensemble loss is a better choice. Even so, AC believes the core idea to introduce the stochasticity to a loss function itself deserves the acceptance. Indeed all reviewers agree on this point. Hence we suggest for the acceptance.

The authors are expected to reflect the rebuttal discussions to the camera-ready version carefully. (Nit: AC is not fully convinced with the argument on the Rademacher complexity as well, but this does not mean a bad thing. To speak of extremes, we could make the loss Lipschitz constant arbitrarily small by multiplicative scaling. Probably we need a better analysis in future)